# Role of UPF1-LIN28A interaction during early differentiation of pluripotent stem cells

Seungwon Jung[1,3], Seung Hwan Ko[1,3], Narae Ahn [1], Jinsam Lee[1], Chang-Hwan Park [1] ✉ & Jungwook Hwang [1,2] ✉

UPF1 and LIN28A are RNA-binding proteins involved in post-transcriptional regulation and stem cell differentiation. Most studies on UPF1 and LIN28A have focused on the molecular mechanisms of differentiated cells and stem cell differentiation, respectively. We reveal that LIN28A directly interacts with UPF1 before UPF1-UPF2 complexing, thereby reducing UPF1 phosphorylation and inhibiting nonsense-mediated mRNA decay (NMD). We identify the interacting domains of UPF1 and LIN28A; moreover, we develop a peptide that impairs UPF1-LIN28A interaction and augments NMD efficiency. Transcriptome analysis of human pluripotent stem cells (hPSCs) confirms that the levels of NMD targets are significantly regulated by both UPF1 and LIN28A. Inhibiting the UPF1-LIN28A interaction using a CPP-conjugated peptide promotes spontaneous differentiation by repressing the pluripotency of hPSCs during proliferation. Furthermore, the UPF1-LIN28A interaction specifically regulates transcripts involved in ectodermal differentiation. Our study reveals that transcriptome regulation via the UPF1-LIN28A interaction in hPSCs determines cell fate.

A premature termination codon (PTC) on mRNA can induce several cellular responses, including nonsense-mediated mRNA decay (NMD), nonsense-mediated alternative splicing (NAS), nonsense-mediated translation repression (NMTR), and nonsense-mediated transcriptional gene silencing (NMTGS)[1–5]. Among PTC-initiating mechanisms, NMD has been exclusively studied to elucidate mRNA decay and translation mechanisms. Mammalian NMD has been proposed for the removal of faulty mRNAs that contain a PTC in its exon to prevent the production of proteins with C-terminal deletions, which could be deleterious to cell metabolism. Many proteins are involved in NMD, including translation termination factors (ribosome, eRF1, and eRF3), kinase for up-frameshift1 (UPF1) (SMG1 and ATM1), exon junction complex (EJC) factors (eIF4AIII, Y14, and MAGOH), EJC-interacting factors (UPF2 and UPF3), proteins interacting with phosphorylated UPF1 (SMG5/7 and SMG6), and mRNA decay factors (XRN1 and DCP1/2)[6–9]. Among various NMD factors, UPF1 is a key factor regulating NMD efficiency that interacts with multiple proteins in its phosphorylated state.

NMD factors have been associated with stem cell proliferation and differentiation. Upf1 is essential for embryonic viability and development, as is seen in zebrafish and *Drosophila*[10–12]. The level of Upf1 is repressed by miR-128, thereby resulting in the downregulation of NMD during murine neuronal differentiation and maintenance of proliferative cell status via TGF-β signalling[13,14]. These observations were confirmed in mouse embryonic stem cells (mESCs)[15,16]. A deficiency of the endonuclease Smg6, which cleaves transcripts after binding to Upf1, inhibits the differentiation of mESCs. In addition, the depletion of NMD factors, including Upf1, delays spontaneous differentiation to embryoid body (EB) formation, suggesting that NMD may be involved in the early stage of differentiation[15].

Another RNA-binding protein, LIN28A, plays a critical role in determining cell fate, including stem cell differentiation and metabolism. The function of LIN28A was first discovered upon identifying a regulatory mechanism in the Lin28/lin-4/let-7 axis during development[17,18]. LIN28A is highly expressed in the early

[1]Graduate School of Biomedical Science & Engineering, Hanyang University, Seoul, Korea. [2]Hanyang Institute of Bioscience and Biotechnology, Hanyang University, Seoul, Korea. [3]These authors contributed equally: Seungwon Jung, Seung Hwan Ko. ✉e-mail: chshpark@hanyang.ac.kr; jwhwang@hanyang.ac.kr

developmental stage, and its expression gradually diminishes during differentiation[19]. Depletion of Lin28a in mESCs reduced the levels of the pluripotency markers Oct4 and Nanog, suggesting that Lin28a is involved in stem cell maintenance[20]. In addition to its function in stem cell maintenance, Lin28 is involved in miRNA maturation from pre-miRNA to miRNA. LIN28A specifically binds to the AAGNNG or AAGNG motifs in a small hairpin, inhibiting specific miRNA maturation and transmembrane protein synthesis[21]. LIN28A is also involved in cellular metabolism. Tetracycline-inducible Lin28a transgenic mice exhibit resistance to obesity and diabetes by promoting an insulin-sensitised state through mTOR signalling[22]. Furthermore, Lin28a enhances mRNA translation for metabolic enzymes, thereby increasing glycolysis and oxidative phosphorylation, resulting in tissue repair[23].

The hypothesis in this study begins with the known roles of UPF1 and LIN28A: (i) proteins with RNA-binding capability, and (ii) involvement in stem cell differentiation. We established that the UPF1-LIN28A interaction impairs NMD and identified the interacting domains of UPF1 and LIN28A. A cell-penetrating peptide (CPP)-conjugated peptide was developed, which specifically disrupted the UPF1-LIN28A interaction and increased NMD efficiency in the presence of LIN28A. Inhibitory peptide induced spontaneous differentiation during proliferation and reduced the abundance of early ectoderm markers during ectoderm-specific differentiation in human pluripotent stem cells (hPSCs). The present study highlights the critical function of the UPF1-LIN28A interaction in hPSCs proliferation and differentiation, which can be modulated by the inhibitory peptide.

## Results

### UPF1 directly interacts with LIN28A

Two post-transcriptional regulators, UPF1 and LIN28A, regulate mRNA stability and stem cell differentiation[19,24,25]. The combination of proteomics analysis and immunoprecipitation (IP) experiments suggested that UPF1 and LIN28A could potentially form a complex in the cell[26,27], which could indicate whether UPF1 and LIN28A would cooperate or compete in regulating mRNA stability during stem cell differentiation. To investigate whether UPF1 and LIN28A interact with each other, FLAG-LIN28A and GST-MYC-UPF1 were expressed in 293T cells, where LIN28A is not constitutively expressed and the expression of exogenous LIN28A was comparable with that of the endogenous LIN28A in hPSCs including hESCs (CHA15 and H9) and human induced pluripotent stem cells (hiPSCs; Pro2). Then, IP was performed in the presence or absence of cellular nucleotides (Fig. 1a and Supplementary Fig. 1a, b). Western blotting (WB) results indicated that FLAG-LIN28A forms a complex with GST-MYC-UPF1, but not with the cap-binding protein eIF4E, suggesting that LIN28A partially interacts with UPF1 in an RNA-independent manner. This result was confirmed by the overexpression of MYC-UPF1 and GST-LIN28A in 293T cells and IP with anti-GST beads (Fig. 1b). Moreover, the UPF1-LIN28A interaction in hPSCs was observed using IP without exogenous expression (Supplementary Fig. 1c). To ensure that these interactions were not due to RNA, DNA, or other proteins, in vitro GST pull-down assay was performed. GST-MYC-UPF1 from 293T cells was purified using GST pull-down, followed by incubation with His-tagged LIN28A. Coomassie blue staining revealed that GST-MYC-UPF1 directly interacted with His-LIN28A, which was confirmed through WB (Fig. 1c, d and Supplementary Fig. 1d). We corroborated the interaction between UPF1 and LIN28A in the cell using confocal microscopy analysis and proximity ligation assay (PLA). Consistent with our findings, confocal microscopy analysis indicated that UPF1 and LIN28A colocalised in hESCs (Supplementary Fig. 1e). As immunostaining assays could not exclude the possibility of indirect interactions, PLA was employed, which allows the observation of two protein interactions within 40 nm (Fig. 1e). Strong signals in embryonic carcinoma cells (PA-1) and hPSCs were observed only in the presence of both anti-UPF1 and anti-LIN28A antibodies, suggesting that UPF1

and LIN28A directly interact within the cells. Finally, we hypothesised that two types of exogenous UPF1 (GST-MYC-UPF1 and HA-UPF1) compete to interact with LIN28A. The coimmunoprecipitated LIN28A with HA-UPF1 was diminished by GST-MYC-UPF1 (Supplementary Fig. 1f). These results indicate that UPF1 partially interacts with LIN28A in an RNA-independent manner.

### LIN28A inhibits NMD

We investigated whether the interaction of LIN28A with UPF1 plays a role in NMD. pFLAG-LIN28A was co-transfected with NMD test plasmids (β-globin [Gl] and glutathione peroxidase 1 [GPx1] with either a normal termination codon or a PTC) (Fig. 2a) and the reference plasmid MUP, which is a control for variation in transfection and RNA recovery. WB and RT-qPCR confirmed that FLAG-LIN28A was efficiently overexpressed, and NMD efficiency (see the definition of NMD efficiency in the Methods section) decreased by approximately 4-fold, suggesting that LIN28A impairs NMD (Fig. 2a). In cases where NMD factors exist downstream of a normal termination codon, the normal termination codon works as a PTC, resulting in mRNA decay[28,29]. Contrary to the removal of Gl mRNA by MS2-HA-UPF1 expression, the amount of Gl mRNA was augmented by an increasing amount of FLAG-LIN28A (Fig. 2b).

Because all evidence that LIN28A attenuates NMD was obtained from the exogenously LIN28A-expressing cells, we employed PA-1 cells, undifferentiated cells[30,31] where LIN28A is expressed to verify that endogenous LIN28A plays a role in NMD. WB results demonstrated that endogenous LIN28A expression was efficiently downregulated, and exogenous FLAG-LIN28A rescued this expression (Fig. 2c). Consistent with the observations displayed in Fig. 2a, b, LIN28A downregulation increased NMD efficiency, which was abrogated by FLAG-LIN28A overexpression. These results were confirmed via tethering experiments using PA-1 cells. Tethered MS2-HA-UPF1 downregulated reporter Gl mRNA, which was more decreased by LIN28A depletion (Fig. 2d). All observations regarding LIN28A impairing NMD were reassessed using RNA sequencing. We performed transcriptome-wide RNA sequencing using the depletion of UPF1 or LIN28A in CHA15 cells. Transcriptome analysis demonstrated that hESCs NMD targets[25] were significantly upregulated upon UPF1-depletion; conversely, NMD targets were significantly repressed upon LIN28A-depletion, suggesting that UPF1 and LIN28A tightly regulated NMD targets in hPSCs (Fig. 2e and Supplementary Fig. 2a). In support, transcriptome analysis in transiently LIN28A-expressing HeLa cells indicated that the expression of endogenous NMD targets[32] in HeLa cells significantly increased in response to LIN28A expression (Supplementary Fig. 2b). Therefore, LIN28A plays a role in NMD.

### LIN28A abrogates UPF1-UPF2 interaction

In an EJC-dependent NMD, eRF3 in a stalled ribosome at a PTC interacts with UPF1 and SMG1, forming a SURF complex (SMG1-UPF1-eRF3), followed by the SMG1-UPF1-UPF2 complex on EJC[6]. Then, mRNA decay is triggered by UPF1 phosphorylation[6,33,34], resulting in the interaction of phosphorylated UPF1 with NMD factors, including the SMG5/7 heterodimer or SMG6[8,35] and the mRNA decay. To decipher the effects of the UPF1-LIN28A interaction on the underlying mechanism, we observed coimmunoprecipitated UPF2 with UPF1 in the presence of LIN28A. The level of UPF2 that coimmunoprecipitated with MYC-UPF1 in GST-LIN28A-overexpressing cells was significantly lower than that in control GST-transfected cells, indicating that the UPF1-LIN28A complex hinders the binding of UPF1 and UPF2 (Fig. 3a). These observations were confirmed in LIN28A-depleted hPSCs, suggesting that the depletion of LIN28A significantly upregulated UPF1-UPF2 interactions (Fig. 3b). As UPF1 phosphorylation occurs after the UPF1-UPF2 interaction, we hypothesised that the increased UPF1-UPF2 interaction due to LIN28A-

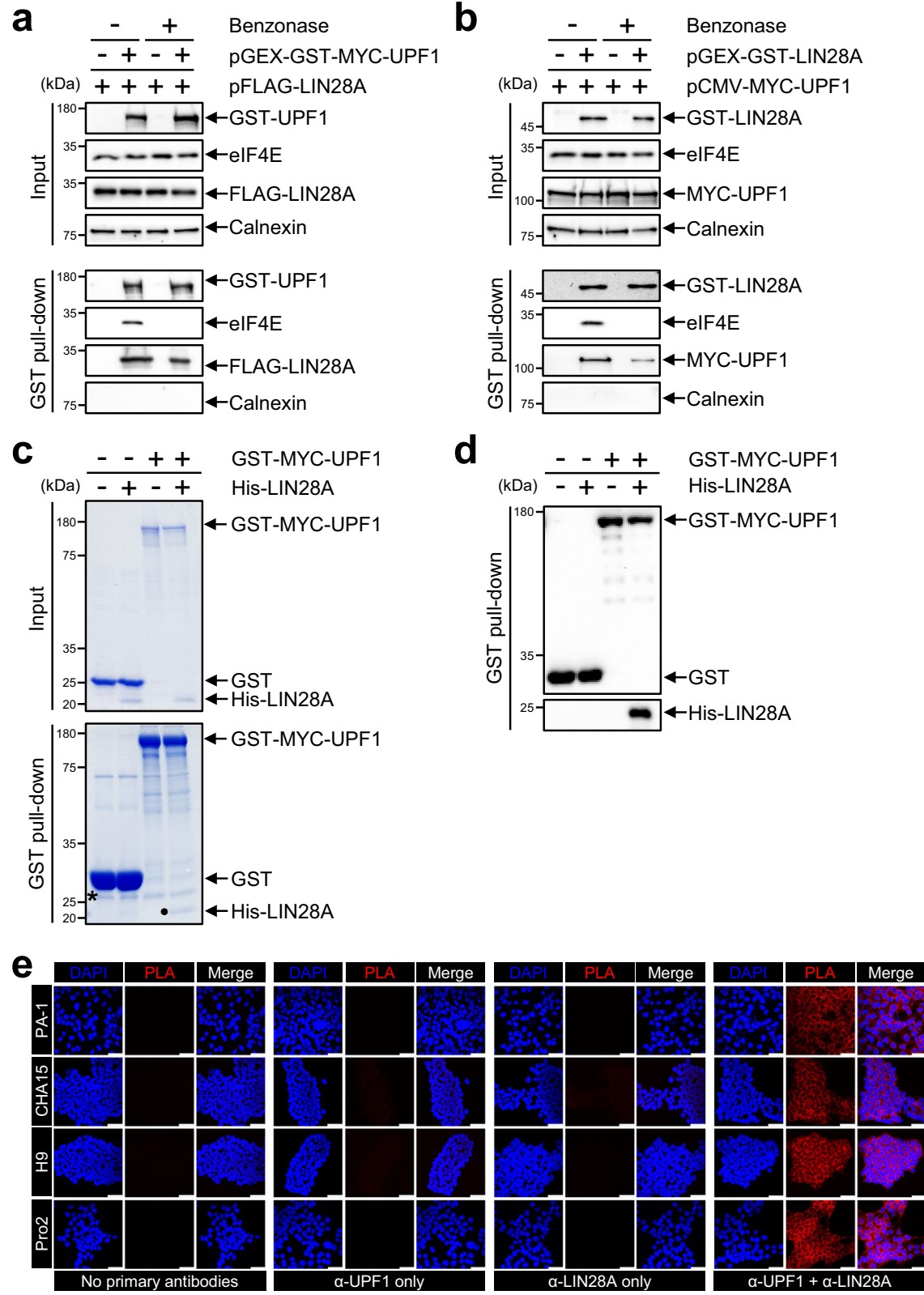

depletion resulted in the upregulation of p-UPF1. The depletion of LIN28A expression in hPSCs increased UPF1 phosphorylation at serine-1096 (S1096) and threonine-28 (T28) (Fig. 3c). Similarly, the UPF1 phosphorylation was reduced by the expression of LIN28A in HeLa cells (Supplementary Fig. 3a). Therefore, the interaction between LIN28A and UPF1 prevents UPF1 phosphorylation by inhibiting the UPF1-UPF2 interaction.

## Generation of inhibitory peptide based on interacting domain of UPF1 and LIN28A

As LIN28A directly interacts with UPF1 and inhibits NMD, inhibiting the UPF1-LIN28A interaction could increase NMD activity. One way to test this hypothesis is to deliver a peptide that exclusively inhibits the interaction between UPF1 and LIN28A, which would be more beneficial than the siRNA-mediated depletion of each protein. To design

**Fig. 1 | UPF1 and LIN28A interact directly. a** 293T cell lysates co-transfected with pFLAG-LIN28A, with pGEX-GST-MYC-UPF1 or pGEX-GST, as the negative control were employed for GST pull-down using GST antibody-conjugated beads in the presence or absence of nuclease (benzonase). **b** Same as (**a**); however, pCMV-MYC-UPF1 were co-transfected with pGEX-GST-LIN28A or pGEX-GST, and a GST pull-down assay was performed. **c** GST-MYC-UPF1 that were purified in GST-MYC-UPF1-expressing 293T cells using GST pull-down in the presence of nuclease mixture was incubated with His-LIN28A followed by GST pull-down. GST pull-down eluates were analysed using Coomassie staining. Dot and asterisk represent His-LIN28A and the unspecific band, respectively. **d** WB was performed to detect GST pull-downed His-LIN28A with GST-MYC-UPF1 in (**c**). **e** PLA in PA-1 and hPSCs, where LIN28A is endogenously expressed, was performed. Two primary antibodies, anti-UPF1 and anti-LIN28A, were recognised using PLA probe-conjugated secondary antibodies. Fluorophore (red) in an amplified DNA circle was observed using confocal microscopy. Blue: DAPI stained nucleus. Scale bars, 50 μm. The minimum number of independent biological replicate experiments was (**a**)–(**e**) $n = 3$. The experiments were conducted three times, each iteration producing consistent results.

inhibitory peptides, we planned to define the domain of UPF1 that interacts with LIN28A. We generated six UPF1 deletional variants including full-length GST-MYC-UPF1 based on the structure and function of the UPF1 domains (Fig. 4a). The cysteine/histidine-rich (CH) domain, which interacts with UPF2, eRF3, and STAU1[6,36–38]. The helicase domain, which comprises the RecA1 and RecA2 domains, functions with the help of ATP binding. GST pull-down assays were performed using cell lysates expressing each UPF1 variant and FLAG-LIN28A. As expected, FLAG-LIN28A was pulled down with full-length UPF1, but not with GST alone. The level of coimmunoprecipitated FLAG-LIN28A was more enriched with GST-MYC-UPF1(419–700) than with other GST-MYC-UPF1 variants (Fig. 4b). Similarly, we identified a UPF1 interacting domain in LIN28A. Six LIN28A functional variants with N-terminal GST-tag were generated (Supplementary Fig. 4a). The levels of coimmunoprecipitated MYC-UPF1 were more enriched with GST-LIN28A(1–77) and GST-LIN28A(78–163) than with GST-LIN28A(164–209) (Supplementary Fig. 4b). Furthermore, we found that the N-terminus of LIN28A interacted with MYC-UPF1 (Supplementary Fig. 4c). Altogether, UPF1 may interact with the cold shock domain in LIN28A, and the interacting domains of UPF1 and LIN28A were mostly conserved in vertebrates (Supplementary Fig. 4d, e).

Based on the interacting domain of UPF1 with LIN28A, we devised a peptide (P1) composed of random amino acid sequences as a negative control and seven peptides (P2–P8) derived from the UPF1 (419–700) domain. Peptides P2–P8 were selected from the surface secondary helix structure in the UPF1 (419–700) domain because therapeutic peptidomimetics blocking protein–protein interaction (PPI) mimics the secondary structure motifs of protein[39] (Fig. 4c). To validate the physical binding to LIN28A of these peptides, we performed surface plasmon resonance (SPR) analysis, where the peptides and His-LIN28A served as analytes and ligand, respectively, suggesting that P8 significantly, and P4 slightly, exhibited a higher binding affinity than others such as a negative control P1 (Supplementary Fig. 4f). Considering that P4 and P8 are closely related to the UPF1 structure, LIN28A may interact with UPF1 via this part of the surface (Fig. 4c). We performed an additional kinetic evaluation to elicit more specific binding characteristics of P8 at diverse concentrations, suggesting that P8 has a high binding affinity to LIN28A (Fig. 4d). Furthermore, mutations of the P8 region in UPF1 (419–700) lost the binding activity with GST-LIN28A (Supplementary Fig. 4g). Taken together, these findings help to identify a peptide that is expected to inhibit the interaction between UPF1 and LIN28A.

**Peptide blocks UPF1-LIN28A interaction**

Next, we investigated whether P8 could hinder the UPF1-LIN28A interaction by competing with endogenous UPF1 for LIN28A. To deliver the peptides into cells, we first generated four N-terminal CPP (N-KKKWCRKKK-C)-conjugated peptides[40] (Fig. 5a). Moreover, amino acid sequences of CPP-P1, P2, and P8 were rarely matched to those from other proteins (Supplementary Data 1). The cell-penetrating efficiency of CPP was investigated by delivering CPP-FITC (Supplementary Fig. 5a). The distribution of CPP-FITC in the cytoplasm verified that the peptide might bind to LIN28A and prevent UPF1-LIN28A interaction in cells where UPF1-LIN28A complexes reside

(Supplementary Fig. 1e and Fig. 1e). To confirm that CPP-P8 could physically inhibit their interaction, each peptide was incubated with cells expressing GST-LIN28A and MYC-UPF1 and a GST pull-down assay was performed (Fig. 5b). The level of coimmunoprecipitated MYC-UPF1 with GST-LIN28A was reduced upon CPP-P8 treatment. Consistent with the observations in Fig. 3c, the disruption of the UPF1-LIN28A interaction by CPP-P8 increased phosphorylated UPF1 in CHA15 and Pro2 cells (Supplementary Fig. 5b), suggesting that the levels of the phosphorylated UPF1 regulated by CPP-P8 may have the observed effect on NMD. To further validate the effect of inhibitory peptide in the context of NMD, we assessed the transcript levels of the NMD reporters in the presence of inhibitory peptide (Fig. 5c). Increased levels of *Gl* Ter and *GPx1* Ter caused by LIN28A expression were efficiently reversed by CPP-P8; however, CPP-P8 itself did not affect NMD in the absence of LIN28A or UPF1 (Supplementary Fig. 5c, d). Finally, PLA was used to confirm the inhibitory function of the peptide in the cells (Fig. 5d). CPP-P8 treatment produced weaker signals than CPP-P1 or -P2, indicating that CPP-P8 efficiently prevented the formation of UPF1-LIN28A complexes in both LIN28A-overexpressing HeLa and CHA15 cells.

**UPF1-LIN28A interaction maintains hPSCs pluripotency**

Once we determined that CPP-P8 interferes UPF1-LIN28A interaction, we investigated the effects of the inhibitory peptide on gene expression in CHA15 cells. Transcriptome analysis using RNA-seq demonstrated that CPP-P8 treatment efficiently depressed the levels of the NMD targets (Fig. 6a). Furthermore, RNA-seqs using UPF1-depleted, LIN28A-depleted, or CPP-P8-treated hPSCs indicated that 230 transcripts including 10 NMD targets were commonly regulated, which were enriched in cellular development and differentiation, suggesting that regulation of NMD targets potentially affects the entire transcriptome (Supplementary Data 2 and 3, Fig. 6b, and Supplementary Fig. 6a). These observations led us to hypothesise that P8 could affect the stemness of hPSCs. When CHA15 cells were cultured in a proliferative state with CPP-P8, the levels of pluripotency markers, OCT4 (encoded by *POU5F1*) and NANOG, were significantly decreased, suggesting that the binding of LIN28A to UPF1 was necessary for pluripotency maintenance (Fig. 6c and Supplementary Fig. 6b). Notably, incubation over 15 days with CPP-P8 in hPSCs altered the cell morphology, induced spontaneous differentiation, and regulated the levels of differentiated makers, indicating that the disruption of UPF1-LIN28A interactions by CPP-P8 has an effect on stem cell maintenance (Supplementary Fig. 6c). The observed transitions may potentially arise from alterations in the transcriptome, as evidenced by RNA-seq analysis conducted on CHA15 cells treated with CPP-P8 for 5 days. This analysis identified 69 transcripts that exhibited a significant two-fold change, and these transcripts were notably enriched in processes related to head development, morphogenesis, pattern specification, and the Wnt signalling pathway (Fig. 6d, Supplementary Fig. 6d, and Supplementary Data 3). Corresponding with the results in Supplementary Fig. 5c, CPP-P8 had less impact on the transcriptome change, including NMD targets, in the absence of LIN28A in hESCs (Supplementary Fig. 6e). In addition, CPP-P8 did not alter the expression of LIN28A-regulating miRNAs and their target mRNAs, suggesting that

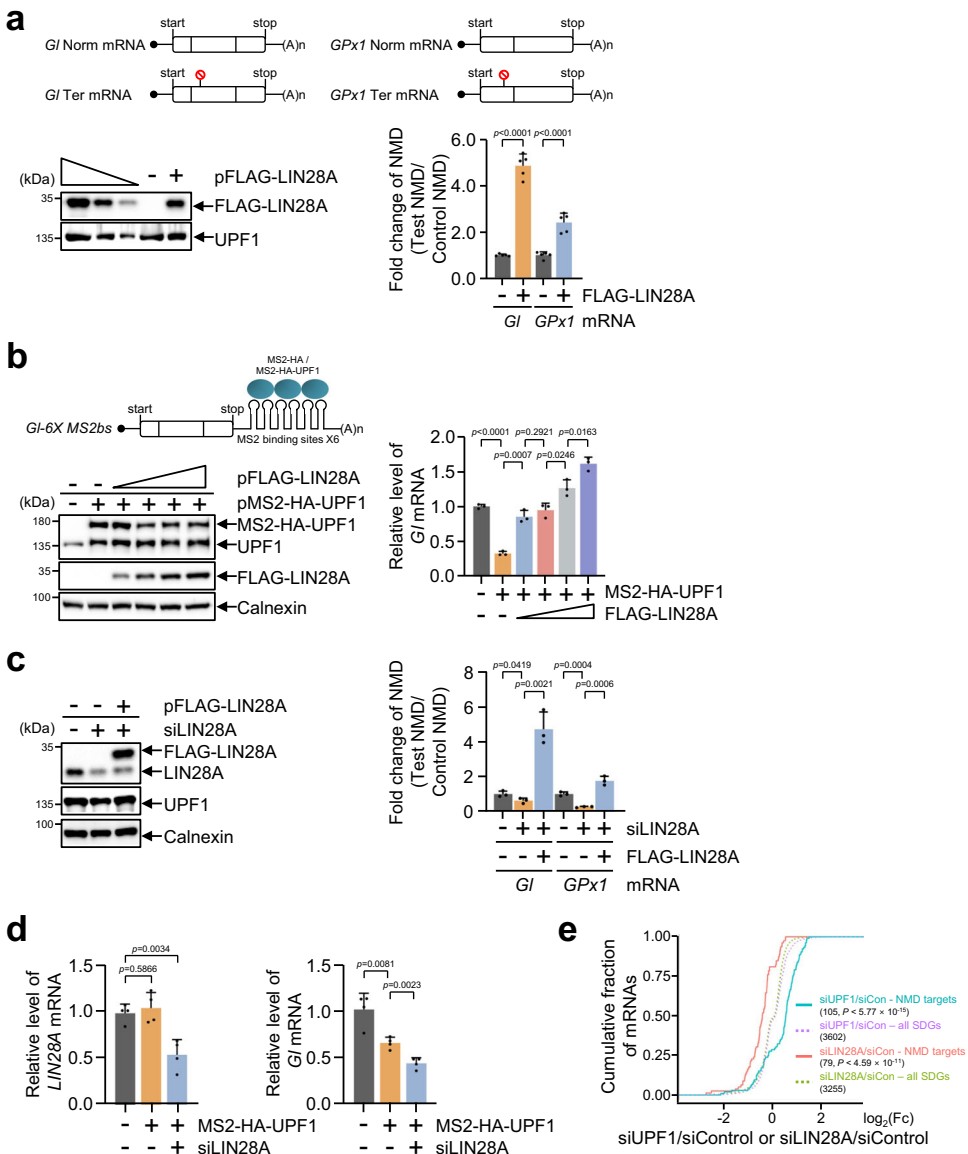

**Fig. 2 | LIN28A impairs NMD. a** Schematic representation of reporter constructs (*Gl* Norm or Ter and *GPx1* Norm or Ter): box, coding region; dots, m⁷Gppp; red stop sign, PTC. HeLa cell lysates co-transfected with pFLAG-LIN28A or pFLAG, with NMD reporter plasmids (Gl and GPx1), were employed for WB and RT-qPCR. **b** Schematic representation of reporter construct (Gl) containing 6X MS2bs downstream of the stop codon for tethering experiment. HeLa cells were co-transfected with four plasmids, including (i) a test plasmid that encodes *Gl* mRNA with six copies of the MS2 coat protein-binding sites within the 3′UTR[28], (ii) an effector plasmid expressing MS2-HA or MS2-HA-UPF1, (iii) gradual increase of pFLAG-LIN28A, and (iv) reference MUP plasmid. WB and RT-qPCR were performed to evaluate the expression of HA-UPF1 and FLAG-LIN28A and the relative amount of *Gl* mRNA, respectively. **c** PA-1 cells were co-transfected with (i) NMD reporter plasmids, (ii)

siLIN28A or control siRNA, and (iii) pFLAG-LIN28A or pFLAG. WB and RT-qPCR were performed to evaluate the expression of LIN28A and NMD efficiency, respectively. **d** Same as in (**b**); however, PA-1 cells were employed in the presence or absence of LIN28A. mRNA levels were normalised to those of *MUP* mRNA in (**a**)−(**d**) and *GAPDH* mRNA in (**d**). **e** Comparative analysis of gene expression profiling using total RNA sequencing from the depletion of UPF1 or LIN28A in CHA15 cells. Cumulative distribution function (CDF) contained significantly differential genes (SDGs, *p* < 0.05) or hESCs NMD targets. Unpaired Student's *t*-test was used for (**a**)−(**d**). Kolmogorov−Smirnov (K−S) tests were used for (**e**). Data are presented as mean values ± SEM. All statistical tests used were two-sided. The minimum number of independent biological replicate experiments was **a** *n* = 5, **b** and **c** *n* = 3, **d** *n* = 4, and **e** *n* = 2.

the transcriptome change by CPP-P8 did not arise from the functions of LIN28A-mediated miRNA regulation (Supplementary Fig. 6f). Therefore, the UPF1-LIN28A interaction modulated hPSCs status via gene regulation.

**Specific inhibition of UPF1-LIN28A interaction affects the expression of ectodermal marker during differentiation**
Given that we confirmed that the inhibitory peptide CPP-P8 blocked UPF1-LIN28A interaction, decreased pluripotent markers, and regulated transcriptomes involved in hPSCs differentiation, it is

reasonable to investigate further whether perturbation of the UPF1-LIN28A interaction was associated with hPSCs differentiation. First, we tracked the expression of *LIN28A*, *POU5F1*, and *UPF1* during three germ-layer differentiation for 12 days (Supplementary Fig. 7a). In contrast to the drastic decrease of *POU5F1* immediately after differentiation, *UPF1* and *LIN28A* levels remained at 50% of their pre-differentiation baseline over five days post-differentiation, indicating that UPF1-LIN28A interaction exists in the early differentiation period. To determine the effects of CPP-P8 during differentiation, we performed RNA-seq using CHA15 cells differentiated into ectodermal,

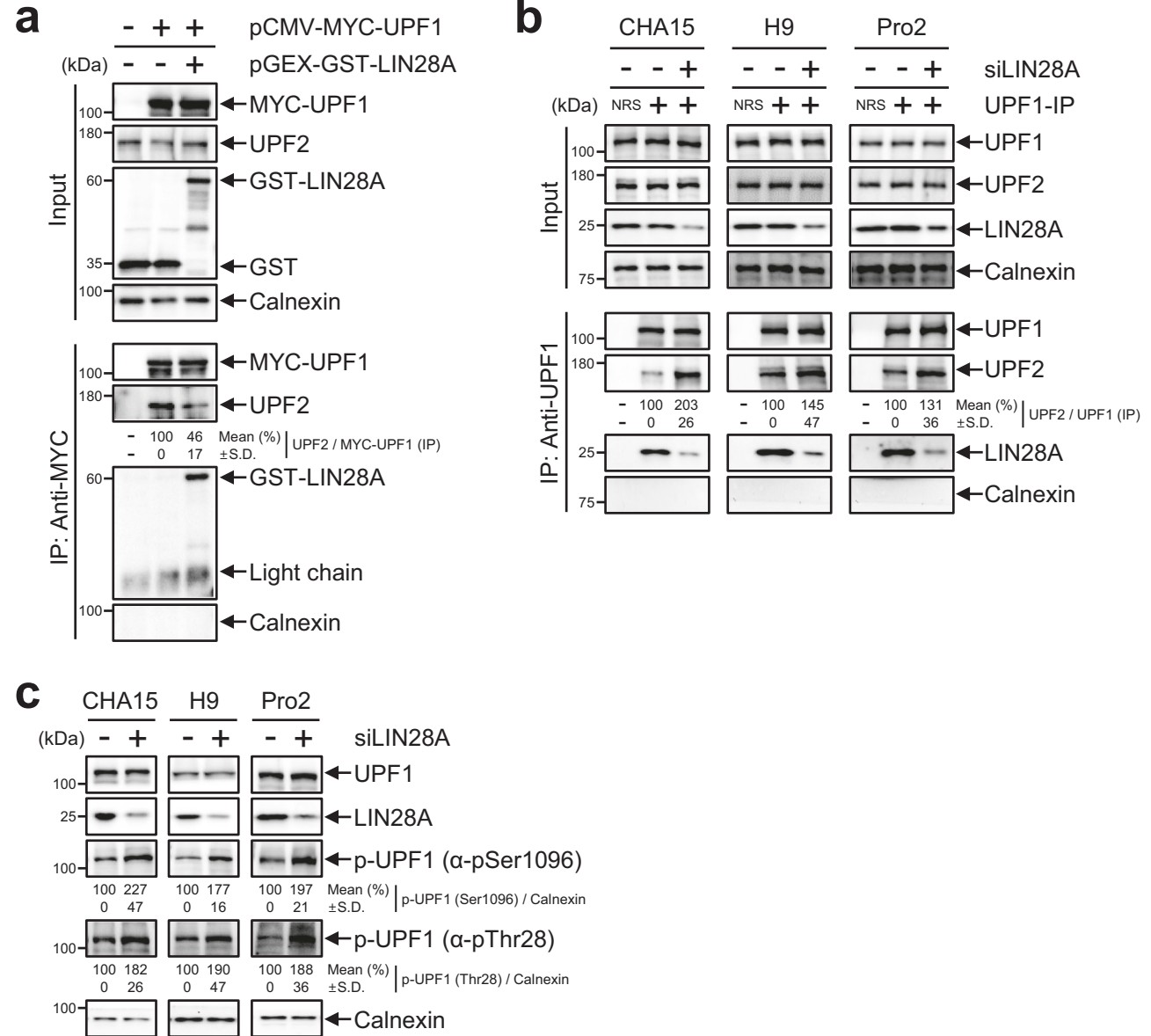

**Fig. 3 | LIN28A diminishes phosphorylated UPF1. a** 293T cells were co-transfected with pCMV-MYC-UPF1 and pGEX-GST-LIN28A followed by IP against anti-MYC beads in the presence of nuclease mixture. **b** Same as (**a**); however, the endogenous UPF1 in hPSCs was immunoprecipitated using UPF1 specific antibody. **c** hPSCs were transfected with siLIN28A or control siRNA. WB was performed to detect phosphorylated UPF1 at serine-1096 or threonine-28. Data are presented as mean values ± SD. All statistical tests used were two-sided. The minimum number of independent biological replicate experiments was **a–c** $n = 3$.

mesodermal, and endodermal lineages in the presence of CPP-P8. Excluding *SOX2* in the ectoderm-differentiated lineage, we observed that the expressions of *NANOG*, *POU5F1*, and *SOX2* were drastically reduced during the differentiation of the three germ layers (Supplementary Fig. 7b). Transcriptome analysis demonstrated that CPP-P8 treatment during ectodermal differentiation overall reduced the levels of its marker, in contrast to mesodermal and endodermal differentiation, suggesting that the UPF1-LIN28A interaction is specifically involved in ectodermal differentiation (Fig. 7a). In the presence of CPP-P8 during ectodermal differentiation, several transcripts were downregulated (*NEUROD1*, *NOS2*, and *PAX3*) or upregulated (*DLK1*, *LHX2*, and *NOTCH1*) (Supplementary Fig. 7c–e). Immunostaining and flow cytometry analysis also indicated that CPP-P8 treatment during ectodermal differentiation abrogated SOX1 expression, but increased DLK1 and LHX2 expression (Fig. 7b and Supplementary Fig. 7f). Therefore, the UPF1-LIN28A interaction is crucial for ectodermal differentiation.

## Discussion

Here, we proved that UPF1 and LIN28A partially interact in an RNA-independent manner to regulate the abundance of natural NMD targets and stem cell maintenance. UPF1 was found to be involved in NMD, mRNA stability control with UPF1 interacting partner proteins[41–43]. However, because most mechanistic studies on UPF1 have been performed using immortal cell lines, the biological significance of UPF1 in hPSCs remains unclear. We proved that the UPF1-LIN28A interaction regulated hPSCs differentiation and fate, and NMD efficiency. We determined the direct interaction between UPF1 and LIN28A via immunoprecipitation and PLA in LIN28A-overexpressing cells and endogenous LIN28A-expressing cells (Fig. 1). These observations were verified by GST pull-down assays using purified proteins. In response to the interaction between UPF1 and LIN28A, the NMD efficiency was repressed (Fig. 2). The level of NMD reporters was destabilised by LIN28A depletion, which was reversed by the expression of LIN28A. In support of this, a tethering assay confirmed that

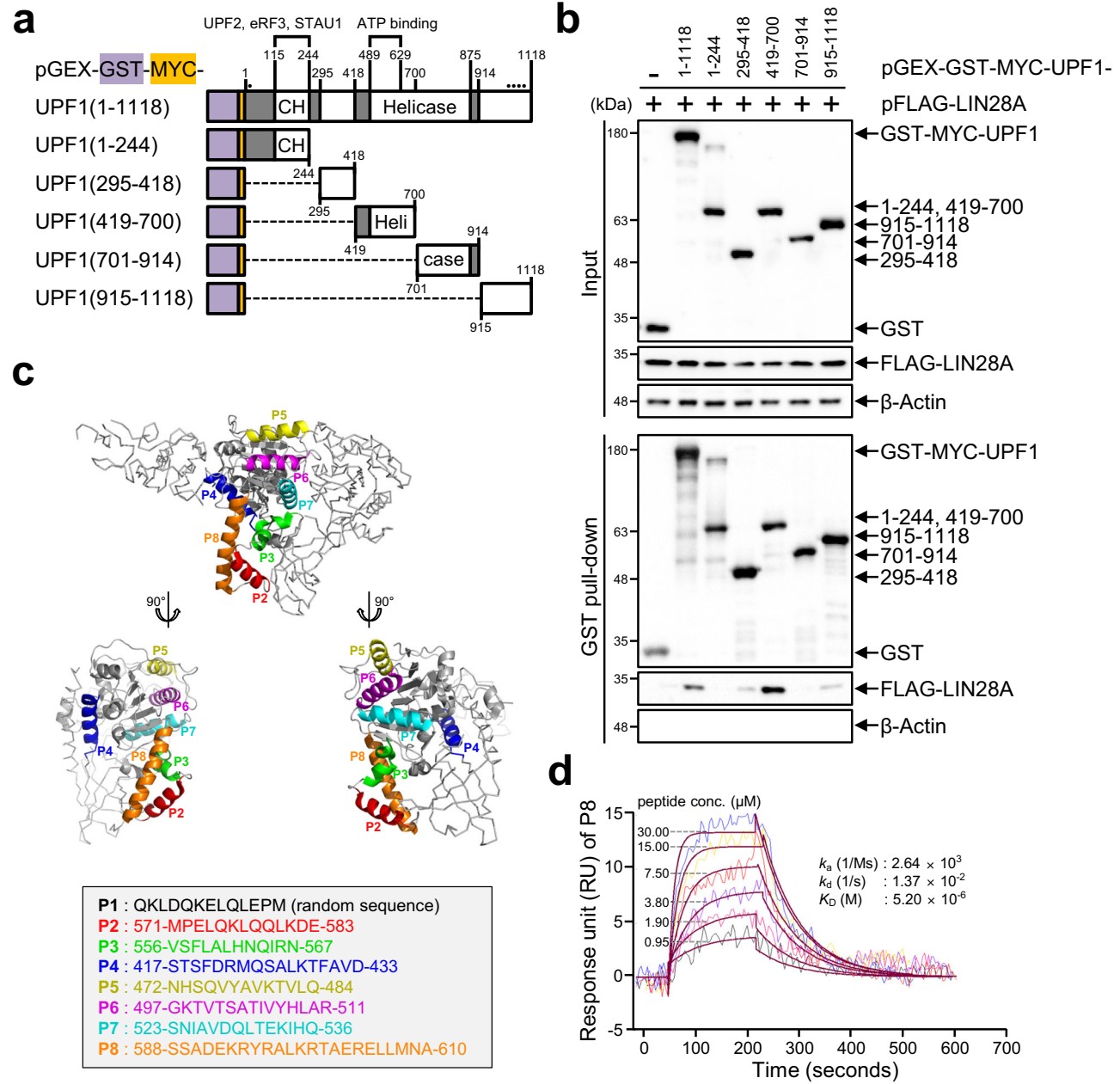

**Fig. 4 | Peptide from UPF1 binds to LIN28A. a** Schematic of deletional GST-MYC-UPF1 variants. CH, Helicase, and SQ represent the cysteine-histidine-rich domain, helicase activity domain, and serine-glutamine-rich domain, respectively. Various proteins interacting with UPF1 domains are indicated. Dots indicate phosphorylation sites on UPF1. **b** Each UPF1 variant in (**a**) and pFLAG-LIN28A was co-transfected in 293T cells followed by GST pull-down in the presence of nuclease mixture. WB was performed to evaluate the expression of deletional variants and eluates. **c** Eight polypeptides, including one random sequence peptide (P1) and seven peptides (P2–P8) from UPF1 (419–700), were depicted with amino acid sequences. Seven peptides (P2–P8) are represented on the UPF1 structure (PDB: 2wjy, 419–700 was displayed with cartoon style). **d** Kinetics evaluations of P8 using SPR analysis were performed at various concentrations. The minimum number of independent biological replicate experiments was **b** $n$ = 3. The experiments were conducted three times, each iteration producing consistent results.

LIN28A stabilised NMD targets, and transcriptome analysis demonstrated that LIN28A expression derepressed endogenous NMD targets.

The blocking of NMD by the interaction of UPF1 and LIN28A seemed to be due to the reduction in UPF1-UPF2 interaction in hPSCs (Fig. 3). NMD is partially dependent on the UPF1-UPF2 interaction[28,44], which leads to UPF1 phosphorylation by SMG1[6]. Notably, LIN28A reduces the interaction between UPF1 and UPF2 by complexing with UPF1, although the interaction motif on UPF1 with LIN28A is structurally separate from that with UPF2 (CH domain on UPF1; Fig. 4b). This could possibly be due to the UPF1-LIN28A interaction being structurally proximal to the CH domain, where UPF2 binds to UPF1[45].

Subsequently, the failure of UPF2 to interact with UPF1 by LIN28A reduces the level of phosphorylated UPF1. Furthermore, we generated UPF1-LIN28A interaction inhibitory peptide that competed with UPF1 for LIN28A (Fig. 4). Delivery of CPP-conjugated peptide effectively suppressed the UPF1 and LIN28A complex, resulting in a change in NMD efficiency (Fig. 5). In support, the disruption of UPF1-LIN28A interaction induced by CPP-P8 for 3 days resulted in an upregulation of p-UPF1 in CHA15 and Pro2 cells, as illustrated in Supplementary Fig. 5b. Interestingly, it is worth noting that CPP-P8 did not elicit a corresponding increase in p-UPF1 levels in H9 cells, even though the depletion of LIN28A was found to upregulate p-UPF1 in this context.

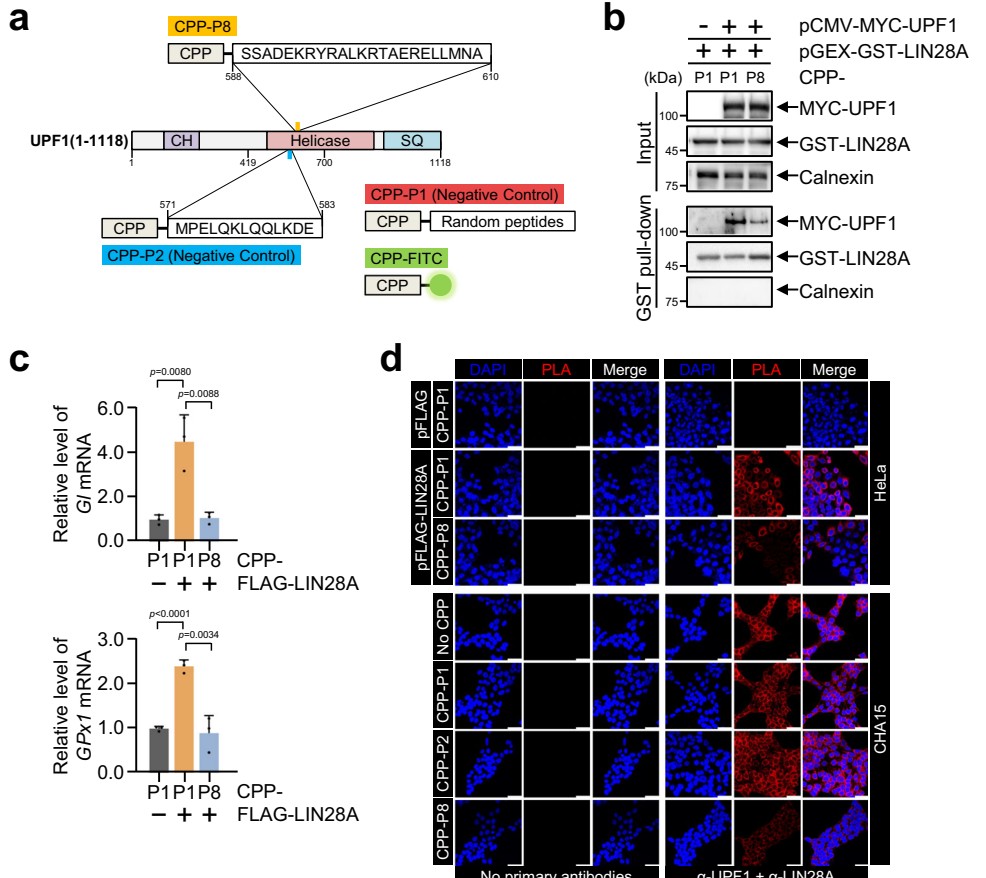

**Fig. 5 | CPP-conjugated peptide inhibits UPF1-LIN28A interaction. a** Schematic representation of designated CPP-conjugated peptides. The amino acid sequences and positions of each peptide on UPF1 are indicated. CPP-P1 and -P2 served as negative controls. CPP-FITC was used to evaluate cell-penetrating ability in cells. **b** 293T cell lysates co-transfected with pCMV-MYC-UPF1 and pGEX-GST-LIN28A and incubated with the indicated CPP-conjugated peptides were employed for GST pull-down. **c** Relative levels of NMD reporters were measured in LIN28A-expressing HeLa cells incubated with the indicated CPP-conjugated peptides. mRNA level was normalised to that of *MUP* mRNA. **d** PLA was performed using LIN28A-overexpressing HeLa cells and hPSCs in the presence of indicated peptides. Red: fluorophore in amplified DNA circle, and Blue: DAPI stained nucleus. Scale bars, 50 μm. Unpaired Student's *t*-test was used. Data are presented as mean values ± SEM. All statistical test used were two-sided. The minimum number of independent biological replicate experiments was **b–d** *n* = 3. The experiments were conducted three times, each iteration producing consistent results.

This observed disparity might be attributed to a temporal delay in the manifestation of CPP-P8's effects. Overall, the direct interaction of UPF1 with LIN28A had a regulatory effect on the transcriptome.

During neuronal cell differentiation, the increased amount of miR-128 repressed UPF1 expression, leading to NMD inhibition[13]. Depletion of Upf1 maintained pluripotency, indicating the involvement of UPF1 in the regulation of proliferation and differentiation within mESCs[15]. Compared with that of UPF1, the role of LIN28A in stem cell differentiation has been studied more extensively[46–48]. LIN28A is highly expressed in hPSCs, similar to that in other stem cell markers such as OCT4, SOX2, and NANOG, and its expression diminishes with differentiation, thereby resulting in LIN28A deficiency in differentiated cells. LIN28 overexpression in hESCs augmented differentiation to a specific endoderm lineage[19]. Therefore, we investigated the biological role of the UPF1-LIN28A interaction in stem cell differentiation. Transcriptome analysis revealed that the depletion of UPF1 and LIN28A in hPSCs upregulated and downregulated NMD targets, respectively, and regulated transcripts categorised in stem cell proliferation and differentiation (Fig. 6). To investigate the function of the UPF1-LIN28A interaction, we employed an inhibitory peptide instead of gene expression using a virus system for two reasons: (i) difficulties in gene expression during differentiation and (ii) that overexpression or depletion of individual gene expression does not explain the effects of specific interaction inhibition. In the context of our proposed model,

wherein the UPF1-LIN28A interaction influences NMD efficiency, leading to transcriptomic alterations, particularly evident in NMD efficiency (Figs. 2e and 6a). The 230 transcripts commonly regulated can be considered as the outcome of the final stage, which is transcriptomic change (Fig. 6b). Thus, although the methods to increase/decrease of NMD efficiency by UPF1-depletion, LIN28A-depletion, and CPP-P8 treatment may differ among the three groups, ultimately they all involve inhibiting UPF1-LIN28A interaction and most regulated transcripts in these three conditions show a similar direction (Supplementary Data 2). The disruption of UPF1-LIN28A interaction by CPP-P8 introduced spontaneous differentiation during proliferation. Notably, disruption of the UPF1-LIN28A interaction modulated the abundance of early ectoderm markers, suggesting that the UPF1-LIN28A complex could regulate early-stage differentiation of hPSCs (Fig. 7). All our observations are summarised in Fig. 7c. Exogenous LIN28A expression in differentiated cells directly interacts with UPF1, inhibiting the formation of UPF1-UPF2 complex and NMD. The endogenous LIN28A-UPF1 complex reduces NMD efficiency in hPSCs, thereby maintaining the stemness of proliferating cells. Especially during ectodermal differentiation, the presence of CPP-P8 downregulated the expression of genes closely associated with CNS development (*DCX*, *NEUROD1*, *NOS2*, and *PAX3*) and upregulated the essential transcription factors for retinal gliogenesis and NOTCH signalling (*LHX2*, *NOTCH1*, and *DLK1*)[49,50] (Supplementary Fig. 7), implying

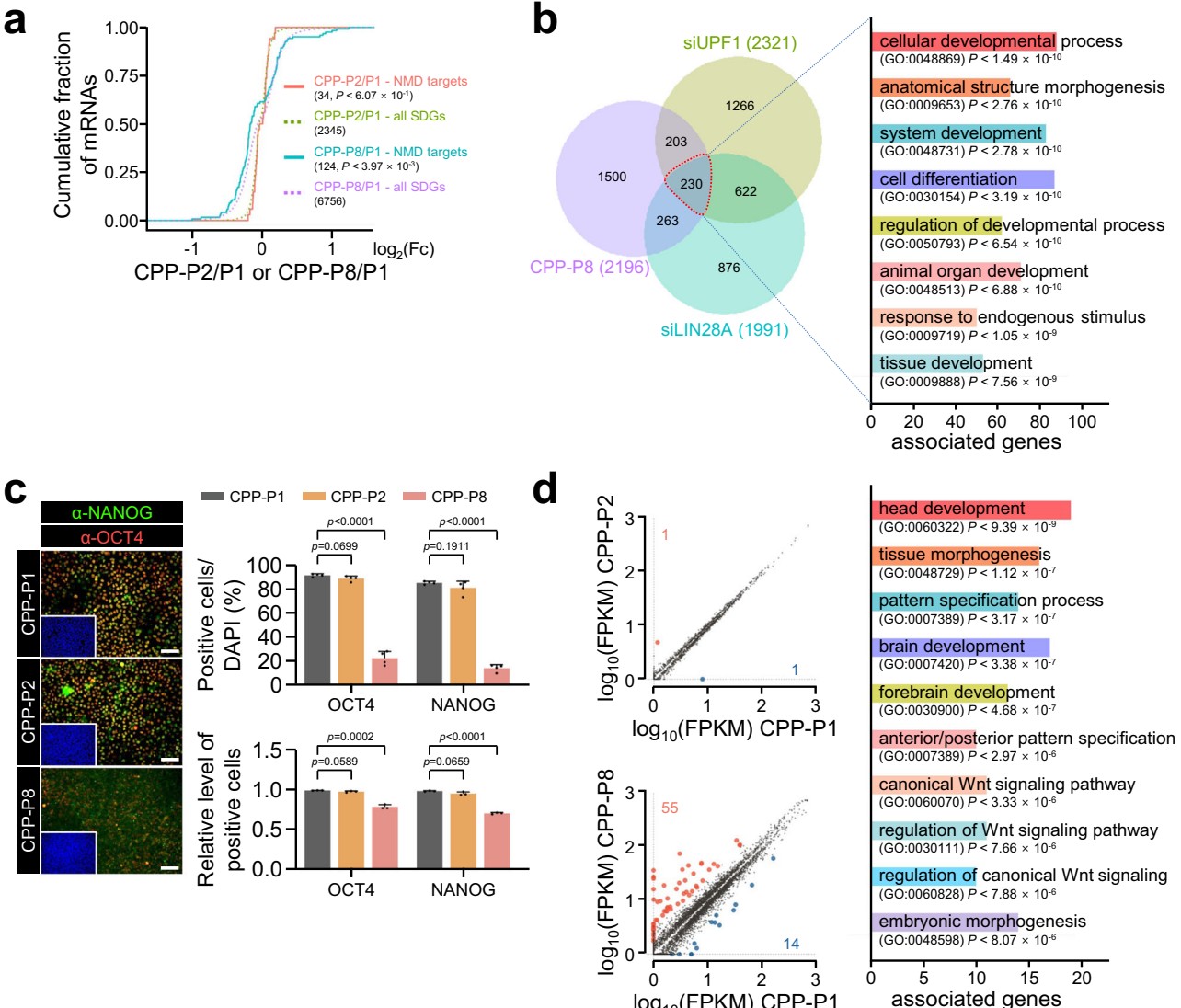

**Fig. 6 | Disruption of UPF1-LIN28A interaction abrogates hPSCs pluripotency via regulation of transcriptome. a** CHA15 cells were treated with 10 μM of the indicated CPP-peptides daily for 5 days. Cumulative fractions of log₂ fold changes in the expression of CPP-P2 or CPP-P8 against CPP-P1 in CHA15 cells were shown as CDF graphs using NMD targets and SDGs. **b** Venn diagram shows the number of transcripts that were significantly changed over 1.2-fold upon UPF1-depletion, LIN28A-depletion, or CPP-P8-treatment in CHA15 cells. The 230 commonly regulated transcripts were used for GO analysis. **c** The expression of OCT4, NANOG, and nuclei were visualised using fluorescence microscopy and the indicated protein-expressing cells were measured using flow cytometry. OCT4- and NANOG-positive cells were counted. Blue: DAPI stained nucleus. Scale bars, 100 μm. **d** Upregulated (red dots) and downregulated (blue dots) transcripts over two-fold significant differential expression (*p* < 0.05) by CPP-P8 or CPP-P2 against CPP-P1 were indicated. Sixty-nine transcripts were employed for GO analysis. K–S tests were used for (**a**). Unpaired Student's *t*-test was used for (**c**). Data are presented as mean values ± SEM. All statistical test used were two-sided. The minimum number of independent biological replicate experiments was **a**, **b**, and **d** *n* = 2, **c** *n* = 3.

that inhibitory peptide could be beneficial for differentiating hPSCs into specific ectodermal tissues related to the retinal gliogenesis differentiation. In this study, we have examined the impact of CPP-P8 on differentiation exclusively in CHA15 cells. Nevertheless, it is crucial to acknowledge that the effects of CPP-P8 on differentiation in H9 and Pro2 cells may yield distinct phenotypic outcomes. This area of investigation remains a subject for future research.

## Methods

### Cell cultivation

HeLa (KCLB, 10002), PA-1 (ATCC, CRL-1572), and 293T (KCLB, 21573) cells were cultured in Dulbecco's modified Eagle's medium (DMEM) supplemented with 10% foetal bovine serum (FBS) and 1% penicillin/streptomycin. Cells were cultured under standard conditions at 37 °C with 5% CO₂.

The hPSCs (CHA-hES15 from CHA Stem Cell Institute, Korea, H9 hESCs from WiCell, and Pro2 iPSCs were kindly gifted by Dr. Kwang-Soo Kim of Harvard University) culture protocol (HYI-17-137-6) was approved by the Institutional Review Board of Hanyang University. hPSCs were cultured in mTeSR Plus Basal Medium with mTeSR Plus 5X Supplement (STEMCELL Technologies, 100-0276) on a Matrigel (Corning, 354277)-coated plate. The culture medium was changed daily, and the cells were passaged using ReLeSR (STEMCELL Technologies, 100-0484). Cells were cultured under standard conditions at 37 °C with 5% CO₂.

### Plasmid construction

To construct pFLAG-LIN28A, the vector pFLAG (Sigma-Aldrich) and PCR-amplified product from *LIN28A* cDNA of PA-1 cells using primers LIN28A_ClaI-F and LIN28A_KpnI-R were digested with *Cla*I (NEB, R0197) and *Kpn*I (NEB, R3142) and ligated.

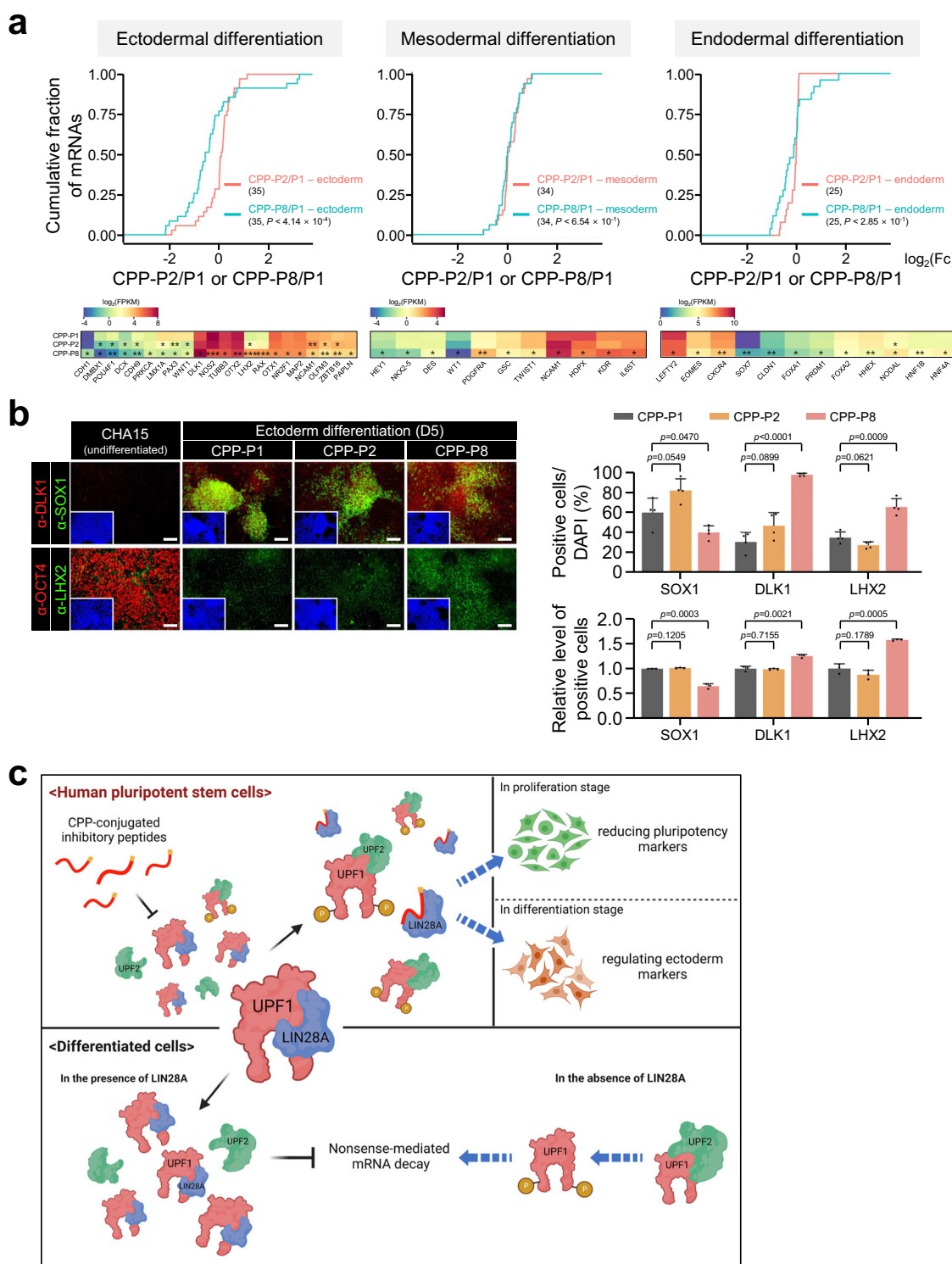

**Fig. 7 | UPF1-LIN28A interaction regulates the expression of ectoderm lineage markers. a** Cumulative fraction of log$_2$ fold change in the expression of the indicated lineage markers upon treatment with CPP-P2 and -P8 against CPP-P1 during ectodermal, mesodermal, and endodermal differentiation of CHA15 cells for 5 days (ectoderm and endoderm) or 4 days (mesoderm) are shown as CDF graphs. Heatmaps represent the relative expressions of significantly differentially expressed lineage markers ($p < 0.05$). **b** Same as in (**a**). Ectodermal differentiation was visualised, along with the expression of pluripotent marker (OCT4), early neural marker (SOX1), and ectodermal makers (DLK1 and LHX2). SOX1-, DLK1-, and LHX2-positive cells were counted. In addition, flow cytometry analysis was performed to evaluate the indicated protein-expressing cells. Blue: DAPI stained nucleus. Scale bars, 100 μm. **c** Suggested model created with BioRender.com. K–S tests were used for (**a**). Unpaired Student's *t*-test was used for (**a**) and (**b**). *$p < 0.05$, **$p < 0.01$, ***$p < 0.001$. Exact *p* values were provided in Source data. Data are presented as mean values ± SEM. All statistical tests used were two-sided. The minimum number of independent biological replicate experiments was **a** $n = 2$, **b** $n = 3$.

To construct GST-tagged LIN28A variants, including full-length (1–209), 1–77, 78–163, 164–209, 1–124, and 125–209, the vector, pGEX-4T-1 (GE Healthcare Life Sciences), and PCR-amplified product from pFLAG-LIN28A using primers LIN28A_XhoI-F and LIN28A_NotI-R for 1–209, LIN28A_XhoI-F and LIN28A(1–77)_NotI-R for 1–77, LIN28A(78–163)_XhoI-F and LIN28A(78–163)_NotI-R for 78–163, LIN28A(164–209)_XhoI-F and LIN28A_NotI-R for 164–209, LIN28A_XhoI-F and LIN28A(1–124)_NotI-R for 1–124, and LIN28A(125–209)_XhoI-F and LIN28A_NotI-R for 125–209 were digested with XhoI (NEB, R0146) and NotI (NEB, R0189) and ligated.

To construct pGEX-GST-MYC-UPF1 variants, including full-length (1–1118), 1–244, 295–418, 419–700, and 701–914, the pGEX-4T-1 vector (GE Healthcare Life Sciences) was digested with SalI (NEB, R0138) and NotI, and the digested vector fragment was ligated to a PCR-amplified product that had been digested with XhoI and NotI. Each PCR-amplified product using primers MYC-UPF1_XhoI-F and MYC-UPF1(1–244)_NotI-R, MYC-UPF1_XhoI-F and MYC-UPF1(295–418)_NotI-R, MYC-UPF1_XhoI-F and MYC-UPF1(419–700)_NotI-R, and MYC-UPF1_XhoI-F and MYC-UPF1(700–914)_NotI-R, was generated using pCMV-MYC-UPF1 variants including 1–1118, 295–914, 419–700, and 701–914, respectively[51].

To construct pGEX-GST-MYC-UPF1(915–1118), the full-length pGEX-GST-MYC-UPF1 and PCR products from the vector were amplified by primers MYC-UPF1(915–1118)_HindIII-F and MYC-UPF1(915–1118)_HindIII-R and digested with HindIII (NEB, R0104) followed by ligation.

To construct pCMV-MYC-UPF1(419–700) mut, each PCR-amplified product using primers UPF1_EcoRI-F and UPF1(588–610)-R and UPF1(588–610)-F and UPF1_NotI-R, was generated using pCMV-MYC-UPF1(419–700). The pCMV-MYC-UPF1(419–700) vector was digested with EcoRI (NEB, R0101) and NotI followed by fusion with two PCR-amplified products with T4 DAN polymerase (Cosmogenetech). All primers used for plasmid construction are listed in Supplementary Data 4.

## CPP-conjugated peptides
Eight peptides were generated (Anygen): P1 (N-QKLDQKELQLEPM-C), P2 (N-MPELQKLQQLKDE-C), P3 (N-VSFLALHNQIRN-C), P4 (N-STSFDRMQSALKTFAVD-C), P5 (N-NHSQVYAVKTVLQ-C), P6 (N-GKTVTSATIVYHLAR-C), P7 (N-SNIAVDQLTEKIHQ-C), and P8 (N-SSA-DEKRYRALKRTAERELLMNA-C). These eight peptides served as analytes in the SPR assay. CPP (N-KKKWCRKKK-C) was conjugated to P1 (CPP-P1), P2 (CPP-P2), P8 (CPP-P8), and FITC (CPP-FITC).

## Surface plasmon resonance (SPR) analysis
For ligand immobilisation, the nitrilotriacetic acid (NTA) surface gold sensor chip (icluebio) was activated with 5 mM $NiCl_2$ (Xantec) at a flow rate of 30 μL/min, following equalisation with 1× HBS-T buffer. His-LIN28A (LS-bio, LS-G637) was used as a ligand protein and was applied to a sensor chip at a flow rate of 10 μL/min for 40 min after dilution in 1× HEPES-buffered saline (HBS-T) buffer at a 1/20 ratio. Approximately 1000 resonance units (RU) were bonded to the sensor chip after ligand immobilisation. RU was measured using iMSPR-ProX (icluebio). An indigenous NTA sensor chip was used as a reference channel.

Analyte-ligand binding screening was performed by tethering the ligand and reference channels in the connected mode. Eight peptides (P1–P8) were dissolved in 1× HBS-T buffer at a concentration of 15 μM. After equalisation with 1× HBS-T buffer, the sensor chip was treated with each peptide for 3 min and then for 6 min with 1× HBS-T buffer for peptide dissociation.

Kinetics assessment was performed for peptides with RU values greater than 5. For kinetic evaluation, analyte-ligand binding screening was performed using concentration titration points at 30, 15, 7.5, 3.8, 1.9, and 0.95 μM for P8.

## siRNA or plasmid DNA transfection
For transient transfection, we seeded $2 \times 10^5$ HeLa, PA-1, or 293T cells into a 6-well plate. After 24 h, we transfected HeLa and PA-1 cells with 300 ng of plasmid DNA and 293T cells with 100 ng using Lipofectamine 3000 (Invitrogen) for HeLa and 293T cells and electroporation (Ingenio) for PA-1 cells. These ratios of cell number and plasmid DNA mass were consistently maintained. In the case of siRNA delivery, we treated hPSCs and PA-1 cells with 100 and 50 nM of siRNA using Lipofectamine 3000, respectively. The synthesised siRNA (IDT technologies) sequences are listed in Supplementary Data 5.

## CPP-conjugated peptide treatment
For CPP-conjugated peptide delivery, HeLa, 293T, and PA-1 cells were incubated with CPP-conjugated peptides or CPP-FITC at 2 μM in DMEM with 10% FBS for 12 h a day before cell harvest. hPSCs were treated with CPP-conjugated peptides at 10 μM in mTeSR Plus Medium, and the peptide-containing medium was changed every day over 10 days. The ROCK inhibitor (Tocris Bioscience, Y-27632) treated-hPSCs were dissociated into single cells using Accutase (Merck, SCR005) and differentiated into ectoderm, mesoderm, and endoderm using the STEMdiff Trilineage Differentiation Kit (STEMCELL Technologies, 05230) following the manufacturer's protocol. Differentiated cells treated with CPP-conjugated peptides were analysed at 0, 3, and 5 days for ectoderm and endoderm, and at 0, 3, and 4 days for mesoderm.

## Total RNA purification and RT-qPCR
Total RNA was extracted using TRIzol reagent (Invitrogen) according to the manufacturer's protocol. After extraction, RNA was incubated with RQ1 RNase-Free DNase (Promega) to remove the exogenous and endogenous DNA.

cDNA was synthesised using RevertAid Reverse Transcriptase (Thermo Fisher Scientific, EP0441) with synthesised random hexamer (Macrogen) for total RNA or stem-loop primers for miRNA. For RT-qPCR reactions, the SensiFAST SYBR No-ROX kit (Meridian Bioscience) was used according to the manufacturer's protocol. All data analysis and visualisation were conducted using R 3.6.1 [www.r-project.org] or GraphPad Prism 10. RT primers for miRNA and RT-qPCR primer sequences are provided in Supplementary Data 6.

## Calculation of NMD efficiency
Fold change in NMD efficiency represents the ratio of test NMD (PTC-containing transcript) to control NMD (PTC-free transcript), where NMD is the relative abundance of PTC-containing mRNA divided by the relative abundance of PTC-free mRNA. The abundance of MUP mRNA was used as the reference.

## RNA sequencing and analysis
Total RNA concentration was calculated using Quant-IT RiboGreen (Invitrogen, R11490). To assess the integrity of the total RNA, samples were run on a TapeStation RNA screentape (Agilent). Only high-quality RNA preparations with RNA integrity number greater than 7.0 were used for the RNA library construction. A library was independently prepared with 1 μg of total RNA from each sample using the Illumina TruSeq Stranded mRNA Sample Prep Kit (Illumina, 20020594). The first step in the workflow involves purifying poly-A-containing mRNA molecules using poly-T-attached magnetic beads. Following purification, mRNA was fragmented into small pieces using divalent cations at elevated temperatures. The cleaved RNA fragments were copied into first-strand cDNA using SuperScript II reverse transcriptase (Invitrogen, 18064022) and random primers. This was followed by second-strand cDNA synthesis using DNA polymerase I, RNase H, and dUTP. These cDNA fragments underwent an end repair process, adding a single 'A' base and ligating the adapters. The products were purified and enriched via PCR to create the final cDNA library. The libraries were quantified using the KAPA Library Quantification Kits for Illumina

Sequencing platforms according to the qPCR Quantification Protocol Guide (KAPA BIOSYSTEMS, KR0405) and qualified using the TapeStation D1000 ScreenTape (Agilent). Indexed libraries were then submitted to an Illumina NovaSeq (Illumina), and Macrogen Inc. performed paired-end (2 × 100 bp) sequencing.

We pre-processed the raw reads from the sequencer to remove low-quality reads and adapter sequences before analysis and aligned the processed reads to *Homo sapiens* (GRCh37) using HISAT v2.1.0[52]. HISAT utilises two indices for alignment (a global whole-genome index and tens of thousands of small local indexes). These two types of indices are constructed using the same Burrows-Wheeler transform (BWT)/graph FM index (GFM) as Bowtie2. The reference genome sequence of *Homo sapiens* (GRCh37) and the annotation data were downloaded from the NCBI database. Transcript assembly was processed using StringTie v1.3.4d[53,54]. Based on this result, the expression abundance of transcripts and genes was calculated as read count or Fragments Per Kilobase of transcript per Million mapped reads (FPKM) value per sample. Differentially expressed genes (DEGs) were analysed by ratio of FPKM or using DESeq2 with read counts[55]. Log$_2$ fold-change value of the NMD targets[25,32], three germ layer markers, miRNA targets, or SDGs was converted into cumulative frequency curve using the R function, 'ecdf' (version 4.0.5). Gene functional classification and Gene ontology (GO) were performed using g:Profiler[56].

### miRNA target selection
Genes regulated by miRNA let-7-5p and miR-16-5p were determined by TargetScanHuman. Among miRNA targets sorted by cumulative weighted context++ score (CWCS)[57], the 100 top genes were selected and analysed using CDF.

### Pull-down assays, in vitro GST pull-down assay, and western blotting
Cells were lysed in ice-cold hypotonic buffer (10 mM Tris-Cl pH 7.5, 10 mM NaCl, 10 mM EDTA, 0.5% Triton X 100, Xpert Protease inhibitor cocktail solution, and Xpert Phosphatase inhibitor cocktail solution (GenDEPOT)). The lysates were centrifuged at 16,000 ×g for 10 min at 4 °C. The supernatants were incubated with protein A or G agarose beads for pre-clearing with a nuclease mixture (DNase I, RNase A, and MNase) or benzonase nuclease (Sigma-Aldrich, 70664; without EDTA) for 1 h at 4 °C. After clearance, centrifugation at 16,000 ×g for 10 min at 4 °C was performed. The supernatants were incubated with anti-MYC, HA, or GST antibody-conjugated beads for 2 h at 4 °C (MYC: Pierce Anti-c-Myc Magnetic Beads (Thermo Scientific, 88842), HA: Pierce Anti-HA Magnetic Beads (Thermo Scientific, 88837), GST: Glutathione Sepharose 4B (Cytiva, GE17-0756-01)). Beads were collected by centrifuging at 3000 ×g for 1 min at 4 °C followed by washing five times with NET2 buffer (50 mM Tris-Cl pH 7.5, 150 mM NaCl, 0.5% NP-40) at 4 °C. The proteins were denatured in a loading buffer containing 5% (v/v) β-mercaptoethanol.

To perform the in vitro GST pull-down assay, the GST pull-down assay for GST-MYC-UPF1 was performed as described. After the bead collecting step, beads were washed five times with NET2 buffer at 4 °C followed by incubation with His-LIN28A protein (LS-bio, LS-G637) in NET2 buffer for 2 h at 4 °C. Beads were collected by centrifuging at 3000 × g for 1 min at 4 °C. After washing five times with NET2 buffer at 4 °C, the proteins were denatured in a loading buffer containing 5% (v/v) β-mercaptoethanol.

Protein samples were loaded for electrophoresis on sodium dodecyl sulphate (SDS) polyacrylamide gels, followed by Coomassie blue staining or transferred to nitrocellulose membranes (Cytiva) for WB. After blocking with 5% non-fat milk (Cellconic) in Tris-buffered saline with Tween-20 for 30 min at room temperature, the membranes were incubated with the following primary antibodies: anti-FLAG (GenScript, A00187; 1:2000), UPF1 (Cell Signaling Technology, 12040; 1:4000), UPF2 (Cell Signaling Technology, 11875; 1:2000), eIF4E (Cell

Signaling Technology, 2067; 1:2000), β-actin (Sigma-Aldrich, A2228; 1:4000), GST (Cytiva, 27-4577-01; 1:2000), MYC (Calbiochem, OP10; 1:2000), Calnexin (Cell Signaling Technology, 2679; 1:4000), LIN28A (Cell Signaling Technology, 3978; 1:2000), pSer1096 phosphorylated UPF1 (1:2000)[58], or p-Thr28 phosphorylated UPF1 (ImmuQuest, IQ653; 1:2000). Primary antibodies were incubated overnight at 4 °C and then washed thrice with TBS-T, followed by incubation with secondary antibodies: Goat anti-Rabbit IgG (Invitrogen, 31460; 1:4000), Goat anti-Mouse IgG (Invitrogen, 31430; 1:4000), or Horse anti-Goat IgG (Vector Laboratories, PI-9500; 1:4000) for 1 h at room temperature. For pull-downed protein detection, all primary and secondary antibodies were diluted at 1:10,000. The blots were imaged using ChemiDoc XRS+ System (Bio-Rad) and the greyscale intensity values were quantified using Image Lab 3.0 software (Bio-Rad).

### Proximity ligation assay
The in situ proximity ligation assay (PLA) was performed on fixed and permeabilised cells using Duolink In Situ Red kit mouse/rabbit (Sigma-Aldrich, DUO92101), according to the manufacturer's protocol. Briefly, fixed and permeabilised cells were incubated with a blocking solution for 1 h at 37 °C and then incubated with the following primary antibodies: anti-UPF1 (Cell Signaling Technology, 12040; 1:200) and LIN28A (Cell Signaling Technology, 5930; 1:200) overnight at 4 °C. After incubation, the coverslips were washed twice with 1× buffer A, followed by incubation with PLA probes for 1 h at 37 °C. After washing twice with 1× buffer A for 5 min, the coverslip was incubated with the ligation solution for 30 min at 37 °C. Coverslips were washed twice with 1× buffer A, followed by an amplification step with polymerase for 100 min at 37 °C. Finally, the coverslip was washed twice with 1× buffer B and once with 0.01× buffer B and then mounted in VECTASHIELD with DAPI mounting medium (Vector Laboratories, H1200).

### Cell staining and microscopy
hPSCs or differentiated cells were fixed in 4% paraformaldehyde (Sigma-Aldrich). After 15–20 min, the cells were washed with 0.1% bovine serum albumin in phosphate-buffered saline (BSA/PBS) thrice and blocked for 1 h by adding 0.3% Triton X-100 (Sigma-Aldrich) and 10% normal goat serum (NGS; Fitzgerald industries, NG22S). After blocking, cells were incubated with the following primary antibodies: anti-UPF1 (Cell Signaling Technology, 12040; 1:500), LIN28A (Cell Signaling Technology, 3978; 1:500), OCT4 (Santa Cruz, sc-5279; 1:500), NANOG (Cell Signaling Technology, 4903; 1:500), DLK1 (GeneTex, GTX60511; 1:500), SOX1 (Cell Signaling Technology, 4194; 1:500), or LHX2 (GeneTex, GTX129241; 1:500). Primary antibodies were incubated overnight at 4 °C followed by incubation with the following secondary antibodies: Cy3 (Jackson ImmunoResearch, 115-165-146; 1:500) or Alexa Fluor 488 (Invitrogen, A-11008; 1:500) followed by the incubation with secondary fluorescence-conjugated antibodies for 1 h. The cells were mounted in VECTASHIELD with DAPI mounting medium (Vector Laboratories, H1200).

Immunofluorescence staining, proximity ligation assay imaging, and cell-penetrating efficiency of CPP-FITC were imaged with a confocal laser scanning microscope (Leica, TSC SP5) or immunofluorescence microscope (Leica, DM5000B). Z-stacks were captured, with sections spanning the entire cell. The LAS X Life Science program was used to obtain the maximum intensity projections and cross-sections of the confocal images.

### Flow cytometry
To evaluate the relative cell number expressing the indicated proteins, CHA15 cells with the indicated CPP-conjugated peptide were fixed with BD Cytofix Fixation Buffer (BD Biosciences Pharmingen, 554655) followed by permeabilization with 0.1% BSA/PBS solution supplemented with 10% NGS and 0.3% Triton X-100 for 1 h at 4 °C. After washing, cells

were incubated with the primary antibodies (OCT4 1:1000, NANOG 1:1000, DLK1 1:1000, SOX2 1:1000, LHX2 1:1000) and then, secondary antibody, Alexa Fluor 488 (1:1000), was applied. The positive cells were determined using flow cytometry (FACSCanto, BD Pharmingen), and data was analysed using the Flow Jo-v10 software program.

### Statistical analysis

Unpaired Student's $t$-tests were used to calculate $p$ values in RT-qPCR experiments, flow cytometry, fluorescence positive cell counting, and DEG in RNA sequencing. Kolmogorov–Smirnov tests were performed for all the CDF analyses. Differences with $*p < 0.05$, $**p < 0.01$, or $***p < 0.001$ were considered significant. Dots in the bar graph indicate the results of individual experiments.

### Reporting summary

Further information on research design is available in the Nature Portfolio Reporting Summary linked to this article.

### Data availability

The data that support this study are available within the article, Supplementary Information, Supplementary Data and Source Data files. Raw RNA-seq data were deposited in the NCBI Gene Expression Omnibus (GEO; https://www.ncbi.nlm.nih.gov/geo/) under accession number GSE224358. Source Data are provided with this paper.

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

## Acknowledgements

This study was supported by the National Research Foundation of Korea (NRF) grants (2020R1A2C2007328 to J.H. and 2019R1A2C2005681 to C.-H.P.), by a grant from the Medical Research Center (2017R1A5A2015395 to J.H.), and by Genome editing research program funded from the Korea government (MSIT) (RS-2023-00261114).

## Author contributions

J.H. and C.-H.P. conceived and devised the study. S.J., S.H.K., N.A., and J.L. performed all experiments. S.J. and S.H.K. performed the bioinformatics and statistical analyses. J.H. and C.-H.P. supervised the study and prepared the manuscript.

## Competing interests

The authors declare no competing interests.
