## [Peer Review File · Nature Communications]

Role of UPF1-LIN28A interaction during early differentiation of pluripotent stem cellsREVIEWER COMMENTS

Reviewer #1 (Remarks to the Author):

The authors show that a critical component of the non-sense mediated decay (NMD) pathway, UPF1, physically interacts with a pluripotent stem cell-specific RNA-binding protein Lin28A. Experiments with NMD test plasmids suggest that this interaction may inhibit the NMD function of UPF1. Consistent with this result, Lin28a also hinders the interaction between UPF1 and another NMD factor, UPF2, and reduces the ability of UPF1 to undergo NMD-specific phosphorylation. The authors then go on to map the UPF1 and Lin28A amino acid sequences mediating the interaction and use this information to design an inhibitory peptide, P8, that that disrupts the interaction. Using a cell-permeable version of P8 the authors show that the UPF1-Lin28A interaction helps human embryonic stem cells (hESC) maintain pluripotency. The authors also believe that this interaction continues to play an important role at early stages of hESC differentiation into the ectodermal but not the mesodermal or the endodermal lineage.

While novel and potentially interesting to a wide readership, the manuscript in its current form lacks some important controls, statistical analyses, and a detailed description of the methods. I also am not fully convinced by the data on the role of Lin28A as an NMD repressor in pluripotent cells and the ectoderm.

Specific comments

1. A central premise of this study is that UPF1 and Lin28A interact directly. To prove this beyond reasonable doubt, the authors should compare, side-by-side, pull-down efficiencies with and without nuclease treatments in Figure 1a. Simply showing that this degrades most of the RNA in Figure S1a is not enough to rule out that the two proteins interact via surviving RNA fragments. A stronger interaction without nucleases would not be necessarily a deal breaker. The two proteins may still interact directly, in addition to binding common RNA fragments. This can be confirmed, for example, by comparing interaction efficiencies between wild-type proteins and their mutants unable to form protein-protein contacts.
2. To ensure that the proximity ligation assay in Fig 1e is specific, the authors should show that the signal with both antibodies is substantially brighter compared to just the UPF1 or the Lin28A antibody.
3. The effect of Lin28A on the UPF1-UPF2 interaction and UPF1 phosphorylation in Figure 3 must be confirmed by repeating the pull-down experiments at least 3 times and quantifying band intensities.
4. I have some concerns about the use of statistics in the manuscript. For example, all the data points in control samples in Figures 2a-d; 3b; S3b,c; and S4a,e,f equal 1, as in no dispersion. This is probably because the authors normalized treatment samples to controls in a pairwise manner. Does such pairwise normalization reflect the design of the experiments? If not, all the data points should be normalized by the control average and the significance of the effects re-tested. Another example is Figure S3b,c where

"ns" is claimed for all samples but +/-siUPF1 samples seem to (and are expected to) differ significantly.

5. Related to the above, the authors do not consistently test statistical significance in their transcriptome-wide analyses. For example, are the up- and down-regulation effects in Figure 6b,d,e, Figure S4b and Supplemental Tables 1 and 3 statistically significant? Are siUPF1-up-regulated genes significantly enriched among siLin28-down-regulated genes in Figure 6b? Do siUPF1 and P8 up- and down-regulate significantly overlapping sets of genes?

6. The authors claim that UPF1-Lin28A interaction is required for ectodermal differentiation. However, Lin28A is enriched in pluripotent stem cells and strongly down-regulated during differentiation. How does this fit authors' model? Is Lin28A expression in differentiating ectoderm sufficient to interfere with UPF1?

Minor points

7. A better description of Results and Methods is needed. For example, the NMD test plasmid experiments are poorly explained and would benefit from simple diagrams of all the constructs in Figure 2, not just GI-6x MS2bs. Plasmid and siRNA concentrations used in transfection assays should be provided throughout the text. The authors should also explain how they classified genes into NMD targets and non-targets. The rationale for depleting ATM1 on page 7 should be explained better. Non-standard abbreviations should be spelled out the first time they are used. Catalog numbers of the key reagents are missing in the Methods section.

8. "UPF1 and Lin28A were mostly conserved in vertebrates (Supplementary Fig. 3d, e)" should refer to Fig. 2d, e.

9. "The inhibitory peptide could be beneficial for differentiating hESCs into specific ectodermal tissues". The authors should explain this point better provided that the inhibitory peptide deregulates ectodermal development.

10. One of the siRNAs in Supplementary Table 5 contains Ts instead of Us. Is this a typo?

Reviewer #2 (Remarks to the Author):

The authors of the manuscript entitled "Role of UPF1-Lin28A interaction during early differentiation of human embryonic stem cells" report an interaction between LIN28 and UPF1 and claim that this interaction regulates the efficiency of NMD. They derive a peptide inhibitor proposed to interfere with the LIN28-UPF1 interaction. They further claim that Lin28-UPF1 interacts to regulate the differentiation potential and pluripotency of hESCs.

The manuscript reports a potentially interesting finding, but, quite bluntly put, neither experiments nor conclusions are up to the standard to be expected from a Nature Communications paper. Many experiments fail to exclude potential artefacts stemming from overexpression of Lin28 (or Upf1). The

expression of RNAseq data is selective and incomplete and clear evidence for a functional role for the interaction of Lin28 and Upf1 is lacking. Crucial data are misinterpreted and there is no evidence for differential UPF1 activity by LIN28 regulation in hESCs. Further, analysis of hESC differentiation potential is crude and I am at a loss to see which conclusions can be drawn from the presented experiments. Overall, the manuscript falls short on providing convincing evidence for a functionally important interaction between Lin28 and Upf1 involved in NMD and pluripotency. It would need considerable extra work and much more careful interpretation of results to perform the experiments needed to test the hypothesis underlying the claims made in this manuscript.

Detailed points:

- The central claim that Lin28 overexpression reduces pUpf1 levels and thereby NMD activity is not supported by the data shown in Fig 3d. Reduced pUpf1 levels apparent on the top two panes follow the reduced actin levels in the bottom pane. This is a crucial experiment that does not support a central claim in the paper.
- In general, it is hard to understand the author's choice of model cell lines. The abstract and title are about human embryonic stem cells, but only a few experiments have been performed in these cells. I have problems understanding why experiments were not performed in hESCs in the first place. hESCs are amenable for genetic modification and biochemistry. Hence, I really can't understand why not choose them for all experiments. As is, there is little evidence that an interaction between Lin28 and Upf1 exists in hES cells.
- Levels of Lin28 overexpression are unclear in Fig2 and many others. It is crucial to test how levels compare with physiological levels? In general, the reliance on overexpression to show an interaction between Upf1 and Lin28a is prone to artefacts – this applies to multiple Figures in the manuscript. The authors made no effort to ensure expression at physiologically meaningful levels. Hence, it is possible that observed interactions by co-IP or PLA are artefacts of overexpression. To show an interaction in a physiologically meaningful context an interaction should be observed in a setting that relies on endogenous levels of protein, and ideally in the cell line (hESCs) that the authors want to study.
- Furthermore, the claim that the proposed Lin28-Upf1 interaction is independent of nucleic acids is not supported by the data. To conclude the Upf-Lin28a interactions are indeed independent of nucleic acids, the authors would need to perform co-IP in cells in their absence, e.g. by performing coIPs in the presence of benzonase or cyanase.
- There is no evidence for the statement “transcripts regulated by UPF1 and Lin28A were involved in the differentiation, development, and migration of stem cells.” I cannot see any specific regulation of stem cell associated genes in Fig 6b. Without any information on the significance at the individual gene level, this analysis remains meaningless.
- The variance for most control measurements is not shown as their values were set to one. This gives the false impression of very robust levels in controls. This needs to be rectified and error bars should also be shown for the controls. The authors are also not clear about how statistical analysis was performed. Did they use the original measured values of the values normalized to one in comparison to controls, if the latter, the results are invalid.
- How is the claim in line 132 that Lin28 preferentially binds to unphosphorylated Upf1 supported by the data. Phosphorylated Upf1 has not been tested in this assay, so no comparison between Upf1 and pUpf1 can be drawn.
- Similarly, reduced coIP of Upf1 with Lin28 after OA treatment is, at best, only circumstantial evidence

for a preferential binding of Lin28 to Upf1.

-The significance of RNAseq data remains unclear. The authors should provide clear data showing statistically significant changes in gene expression. How many individual NMD targets (or stem cell associated genes) show a significantly altered expression profile? How many other genes show a significant change. Is the enrichment of NMD targets significant? How exactly was ensured that the selection of random genes does not produce an artefact? Have multiple selection of 284 genes been performed in a bootstrapping approach to make sure that the choice of the author's 284 random genes was not a random 'lucky' pick? If so, how many permutations have been performed? Did the authors ensure that the random selected genes are expressed to a similar level as proposed NMD targets? The tables provided are not helpful and clearly below the expected reporting standard.

-How can the authors be sure that the effect of the Upf1 P8 peptide is specifically blocking the interaction with Lin28? Isn't it possible that the same domain serves as blocking site for other interactors as well?

-The effect of CPP8 on hESCs is not characterized well enough. Indeed, Nanog expression is lost upon treatment, but what is the fate of these cells? Will they die (as expected after inhibition of Upf1 function)? In line with this, what is the fate of hESCs upon deletion of Lin28a and Upf1? Are these cells still ESCs?

-Are any of the transcriptional changes in the heatmaps shown in Fig 8a statistically significant?

-In Fig2b increased Lin28 levels coincide with reduced MS2-Upf1 levels. Did the authors take into account that these reduced Upf levels could contribute to reduced NMD activity?

Minor:

-the authors state that all Lin28 OE experiments were performed in Lin28 KO cells. How do NMD levels in Lin28 KO cells compare to WT cells?

-How do transcript profiles of CPP-P8 treatment and UPF1 or LIN28 KD correlate?

Reviewer #3 (Remarks to the Author):

As an important RNA monitoring mechanism in eukaryotic cells, nonsense-mediated mRNA decay (NMD) identifies and degrades mRNAs containing premature termination codons (PTCs) in the open reading frame, which is important for maintaining orderly genes expression system. This research identifies the interaction of LIN28A and UPF1 (an important regulator of NMD) and the effect of this complex on NMD. The authors developed a cell-penetrating peptide (CPP) based on the LIN28A-UPF1 interaction structure to inhibit the interaction between LIN28A and UPF1. The addition of CPP-8 in hESC effectively inhibited the binding of LIN28A to UPF1, regulated the expression of ectodermal markers during differentiation, and disrupted hESC pluripotency.

Major Points:

1. In Figure 1, although different exogenous protein methods were used in different cells to examine the

interactions between LIN28A and UPF1 in an RNA-independent manner, it is strongly recommended to validate LIN28A-UPF1 interactions in hESCs with immunoprecipitation of endogenous proteins, and comparing the effects before and after RNAase addition.

2. A variety of cell lines, including PA-1, HeLa, and 293T, were used in the study to demonstrate the regulatory effects of LIN28A and UPF1 on NMD. However, it is likely that hESC lines have different contexts and mechanisms. Many of the molecular experiments in this paper were not repeated in 3 different hESC cell lines (Figure 2-3). Another possibility to consider is iPSCs.

3. Conventional immunofluorescence assay is affected by exposure time, focal length, etc., which makes it unsuitable for quantitative analysis. The authors should use an advanced instrument that can be used for quantitative analysis (Figure 6f, 7b).

4. In addition to identifying the differentiation markers, what are the effects of LIN28A-UPF1 interactions on the functional differentiation phenotypes?

Minor points:

5. Results for quantitative analysis by Western Blot are recommended to be labeled with grayscale intensity values above the protein bands (e.g. Figure 2).

6. The RT-qPCR results suffer from a problem of reproducibility and variability (especially supplementary Figure 5).

7. Many of the gene names, mRNA names, and protein names in the Figures do not follow the standard conventions.

We have performed additional experiments and statistical analyses that were suggested by the reviewers and have revised the manuscript to address the reviewers' concerns. Furthermore, (1) we have re-performed biochemical experiments and reproduced all results using different human pluripotent stem cells (hPSCs) including human embryonic stem cells (hESCs; CHA15, and H9) and human induced pluripotent stem cells (hiPSC; Pro2). (2) To strengthen the evidence for the direct interaction between UPF1 and LIN28A, we employed more stringent conditions using benzonase nuclease in both immortalized cell lines and hPSCs.

REVIEWER COMMENTS

Reviewer #1 (Remarks to the Author):

The authors show that a critical component of the non-sense mediated decay (NMD) pathway, UPF1, physically interacts with a pluripotent stem cell-specific RNA-binding protein Lin28A. Experiments with NMD test plasmids suggest that this interaction may inhibit the NMD function of UPF1. Consistent with this result, Lin28a also hinders the interaction between UPF1 and another NMD factor, UPF2, and reduces the ability of UPF1 to undergo NMD-specific phosphorylation. The authors then go on to map the UPF1 and Lin28A amino acid sequences mediating the interaction and use this information to design an inhibitory peptide, P8, that that disrupts the interaction. Using a cell-permeable version of P8 the authors show that the UPF1-Lin28A interaction helps human embryonic stem cells (hESC) maintain pluripotency. The authors also believe that this interaction continues to play an important role at early stages of hESC differentiation into the ectodermal but not the mesodermal or the endodermal lineage.

While novel and potentially interesting to a wide readership, the manuscript in its current form lacks some important controls, statistical analyses, and a detailed description of the methods. I also am not fully convinced by the data on the role of Lin28A as an NMD repressor in pluripotent cells and the ectoderm.

Specific comments

1. A central premise of this study is that UPF1 and Lin28A interact directly. To prove this beyond reasonable doubt, the authors should compare, side-by-side, pull-down efficiencies with and without nuclease treatments in Figure 1a. Simply showing that this degrades most of the RNA in Figure S1a is not enough to rule out that the two proteins interact via surviving RNA fragments. A stronger interaction without nucleases would not be necessarily a deal breaker. The two proteins may still interact directly, in addition to binding common RNA fragments. This can be confirmed, for example, by comparing interaction efficiencies between wild-type proteins and their mutants unable to form protein-protein contacts.

Response: Thank you for the valuable comments. As suggested, we tested the interaction of UPF1 and LIN28A in the presence or absence of benzonase nuclease (more stringent nuclease as suggested by Reviewer #2) by immunoprecipitation (IP) in hPSCs and LIN28A-overexpressing 293T cells. IP and western blotting (WB) demonstrated that UPF1 partially binds to LIN28A in an RNA-independent manner (revised manuscript Fig. 1a, 1b, and Supplementary Fig. 1c). These results were expected because both proteins are RNA-binding proteins and indirect interaction of UPF1 and LIN28A may be coimmunoprecipitated through RNA. To strengthen the direct binding, we employed UPF1 mutation on P8 peptide in UPF1(419-700), which binds to LIN28A in an RNA-independent manner. WB results demonstrated that mutation on P8 in UPF1(419-700) lost LIN28A binding activity in the presence of benzonase, suggesting that UPF1 directly binds to LIN28A (Supplementary Fig. 4g in the revised manuscript). Thus, we asserted that UPF1 binds directly to LIN28A and provide several evidences: i) *in vitro* binding assay using purified proteins (Fig. 1c and 1d in the revised manuscript), ii) IP assay in the presence of stringent nuclease (Fig. 1a, 1b, and Supplementary Fig. 1c in the revised manuscript), iii)

PLA assays (Fig. 1e in the revised manuscript), iv) deletion analysis using IP (Fig. 4b, Supplementary Fig. 4b, and 4c in the revised manuscript), and v) mutational assay on UPF1 using IP (Supplementary Fig. 4g in the revised manuscript). We hope these additional experimental results and explanations adequately address the reviewer's concerns.

2. To ensure that the proximity ligation assay in Fig 1e is specific, the authors should show that the signal with both antibodies is substantially brighter compared to just the UPF1 or the Lin28A antibody.
 Response: Thank you for the constructive comment. As the reviewer suggested, we reformed PLA assay in the presence of i) UPF1 antibody, ii) LIN28A antibody, and iii) both UPF1 and LIN28A antibodies in hPSCs, as well as PA-1 cells, all of which constitutively express LIN28A. Confocal microscopy demonstrated that the fluorophore of DNA was only observed in the presence of both antibodies, suggesting that UPF1 and LIN28A directly interact within the cells (Fig. 1e in the revised manuscript). We hope these results address the reviewer's concerns.

3. The effect of Lin28A on the UPF1-UPF2 interaction and UPF1 phosphorylation in Figure 3 must be confirmed by repeating the pull-down experiments at least 3 times and quantifying band intensities.

Response: Thank you for the suggestion. Accordingly, we have repeated the UPF1 IP experiment using an anti-MYC antibody in Fig. 3a and quantified the colPed UPF2 (Fig. 3a in the revised manuscript). The level of colPed UPF2 with MYC-UPF1 in the presence of GST-LIN28A reduced to approximately 2-fold compared with the one of colPed in the control.

As we employed transiently MYC-UPF1-overexpressing cell lines in Fig. 3a, we examined UPF1-UPF2 interaction by UPF1 IP with nuclease treatment in LIN28A-depleted hPSCs (requested by Reviewer 3, Fig. 3b in the revised manuscript). IP and WB results demonstrated that the level of colPed UPF2 with UPF1 increased up to approximately 1.5~2.0-fold upon LIN28A-depletion.

In addition to repeating Fig. 3a, we verified the upregulation of phosphorylated UPF1 by depletion of LIN28A in hPSCs. We depleted the endogenous LIN28A in hPSCs and quantified the increasing level of the phosphorylated UPF1 with statistical analysis, suggesting that LIN28A is involved in the UPF1 phosphorylation status (Fig. 3c in the revised manuscript). For the final additional experiment, we treated hPSCs with CPP-P8 and observed the p-UPF1 levels in hPSCs. Our findings reveal that the disruption of UPF1-LIN28A interaction induced by CPP-P8 for 3 days resulted in an upregulation of p-UPF1 in CHA15 and Pro2 cells, as illustrated in Supplementary Figure 5b. Interestingly, it is worth noting that CPP-P8 did not elicit a corresponding increase in p-UPF1 levels in H9 cells, even though the depletion of LIN28A was found to upregulate p-UPF1 in this context. This observed disparity might be attributed to a temporal delay in the manifestation of CPP-P8's effects. We hope that the reviewer finds that our results, namely that UPF1-UPF2 interaction and UPF1 phosphorylation status depends on LIN28A, are now acceptable.

Not available

Fig. 3d

Fig. 3c

Supplementary Fig. 5b

Not available

4. I have some concerns about the use of statistics in the manuscript. For example, all the data points in control samples in Figures 2a-d; 3b; S3b,c; and S4a,e,f equal 1, as in no dispersion. This is probably because the authors normalized treatment samples to controls in a pairwise manner. Does such pairwise normalization reflect the design of the experiments? If not, all the data points should be normalized by the control average and the significance of the effects re-tested. Another example is Figure S3b,c where "ns" is claimed for all samples but +/-siUPF1 samples seem to (and are expected to) differ significantly.

Response: We agree with this reviewer regarding statistical analysis, which was also raised by Reviewer #2. We reanalyzed all the data as the reviewer suggested (normalization by the control average and the significance were recalculated) and replaced the original figures with the revised ones (We displayed the one example as below). Regarding "ns" in Supplementary Fig. 3c in the original manuscript, we did make a mistake in the description. "ns" should be the data among siUPF1 samples, not between +/-siUPF1 samples. We apologize for our mistake and replace the figures as below. We hope this explanation suitably addresses your concerns.

Original Manuscript

Revised Manuscript

Fig. 2d

Fig. 2d

Supplementary Fig. 3c

Supplementary Fig. 5d

5. Related to the above, the authors do not consistently test statistical significance in their transcriptome-wide analyses. For example, are the up- and down-regulation effects in Figure 6b,d,e, Figure S4b and Supplemental Tables 1 and 3 statistically significant? Are siUPF1-up-regulated genes significantly enriched among siLin28-down-regulated genes in Figure 6b? Do siUPF1 and P8 up- and down-regulate significantly overlapping sets of genes?

Response: We appreciate this comment. The reviewer's concern is similar to the one raised by Reviewer #2. To obtain statistical significance, we repeated i) UPF1- or LIN28A-depletion in hPSCs (corresponding to Fig. 6a and 6b, Supplementary Fig. 4b, and Supplementary Table 1 in the original manuscript), ii) peptide treatment in hPSCs (corresponding to Fig. 6c, 6d, and 6e, and Supplementary Table 3 in the original manuscript), and iii) peptide treatment during three germ layer differentiation (corresponding to 7a in the original manuscript). We reperfomed RNA-seqs and reanalyzed the data using significantly changed genes ($p < 0.05$) and obtained statistical significance.

After speculation, we believe the results in Fig. 6b in the original manuscript did not provide much information regarding NMD targets and stem cell-associated transcripts. Instead, we first provided overlapping gene sets using significantly changed transcripts ($|F_c| > 1.2$, $p < 0.05$) upon UPF1-depletion, LIN28A-depletion, or P8-treatment in hPSCs (Fig. 6b in the revised manuscript). Comparative analysis using RNA-seq results revealed that 2321, 1991, and 2196 transcripts were significantly regulated upon UPF1-, LIN28A-depletion, and P8-treatment, respectively, and 230 transcripts were commonly regulated. Thereafter, we performed gene ontology (GO) enrichment analysis using the commonly regulated 230 transcripts (Fig. 6b in the revised manuscript), indicating that transcripts regulated by UPF1, LIN28A, and CPP-P8 were involved in the cellular development and differentiation. In addition, we added the significance values with fold-change in Supplementary Table 2 in the revised manuscript. To address the levels of NMD targets, which is supposed to be upregulated and downregulated by UPF1- and LIN28A-depletion, respectively, we performed RT-qPCR to measure NMD targets using biologically replicated hPSCs samples ($n=3$) (Supplementary Fig. 2a in the revised manuscript). RT-qPCR results demonstrated that NMD targets in hPSCs were significantly upregulated and downregulated upon UPF1-depletion and LIN28A-depletion, respectively.

Regarding gene sets regulated by UPF1 and CPP-P8, we performed transcriptome analysis using significantly changed NMD targets ($p < 0.05$) in UPF1-depleted or CPP-P8 treated CHA15. Of these, 75 and 76 transcripts were upregulated and downregulated upon UPF1-depletion and CPP-P8 treatment, respectively (below figure, reviewer only version) and there were 18 common transcripts were, suggesting that approximately 24% transcripts overlapped.

We hope that the reviewer will find that these revised results strongly support our statistical analysis.

Original Manuscript

Revised Manuscript

Fig. 6b

Not available

Fig. 6d

Fig. 6d

Fig. 6e

Fig. 6b

Supplementary Fig. 4b

Not available

Supplementary Fig. 2a

Not available

Reviewer only

Not available

6. The authors claim that UPF1-Lin28A interaction is required for ectodermal differentiation. However, Lin28A is enriched in pluripotent stem cells and strongly down-regulated during differentiation. How does this fit authors' model? Is Lin28A expression in differentiating ectoderm sufficient to interfere with UPF1?

Response: We agree with the reviewer's concerns. As noted, LIN28A is highly expressed in hPSCs and its expression is gradually decreased during differentiation. To determine the levels of *LIN28A* during differentiation, we measured the expression of *LIN28A*, *UPF1*, and *POU5F1* (OCT4) mRNA during three germ layer differentiation for 12-days (Supplementary Fig. 7a in the revised manuscript). RT-qPCR results demonstrated that the levels of *POU5F1* dramatically reduced immediately after differentiation. However, 5 days after differentiation (we performed RNA-seq at 5-day differentiation in ectodermal and endodermal differentiation and, at 4-day differentiation in mesodermal differentiation), the levels of *LIN28A* and *UPF1* remained over 50% at a minimum. Furthermore, the FPKM of *LIN28A* during differentiation was more than that of *UPF1* by approximately 7~28-fold depending on differentiation layer, suggesting that *LIN28A* expression could be higher than *UPF1* expression. The simple interpretation of these results indicated that *LIN28A* expression in differentiation could be enough to interact with *UPF1*, although *LIN28A* expression decreased during differentiation. We trust that this explanation will now satisfy these concerns.

Original Manuscript

Revised Manuscript

Supplementary Fig. 7a

Not available

Minor points

7. A better description of Results and Methods is needed. For example, the NMD test plasmid experiments are poorly explained and would benefit from simple diagrams of all the constructs in Figure 2, not just GI-6x MS2bs. Plasmid and siRNA concentrations used in transfection assays should be provided throughout the text. The authors should also explain how they classified genes into NMD targets and non-targets. The rationale for depleting ATM1 on page 7 should be explained better. Non-standard abbreviations should be spelled out the first time they are used. Catalog numbers of the key reagents are missing in the Methods section.

Response: Thank you for the valuable suggestions. We included the schematic representation of NMD reporter plasmids in Fig. 2a in the revised manuscript. Furthermore, we added the amount and concentrations of DNA plasmid and siRNA used in the revised manuscript. Particularly, the amount of plasmid DNA expressing FLAG-LIN28A was determined to express the comparable levels of the endogenous LIN28A in hPSCs (Supplementary Fig. 1b in the revised manuscript). Regarding NMD targets, we employed previously reported NMD targets^{1,2}. Because NMD targets in HeLa cells and hPSCs are different from each other, we employed the different NMD target datasets when specified.

As we now removed Fig. 3b and 3c in the original manuscript to improve the manuscript and make it more rational, which was addressed in question #7 raised by Reviewer 2, we would like to skip this particular concern. Furthermore, we added the full name before the abbreviation and catalog numbers of reagents. We hope the additional information is acceptable.

Original Manuscript

Revised Manuscript

Fig. 2a

Fig. 2a

Supplementary Fig. 1b

Not available

8. "UPF1 and Lin28A were mostly conserved in vertebrates (Supplementary Fig. 3d, e)" should refer to Fig. 2d, e.

Response: We apologize for our mistake. We corrected the reference accordingly.

9. "The inhibitory peptide could be beneficial for differentiating hESCs into specific ectodermal tissues".

The authors should explain this point better provided that the inhibitory peptide deregulates ectodermal development.

Response: When the inhibitory peptide was administered under ectodermal differentiation, the expression of genes that are closely associated with CNS development, such as *DCX* and *NEUROD1*, decreased, and overall ectodermal differentiation was inhibited. *LHX2*, an essential transcription factor for retinal gliogenesis and NOTCH signaling^{3, 4}, was significantly increased, indicating that our inhibitory peptide may promote early eye development. Accordingly, we modified the statement following

Before: “Inhibitory peptides could be beneficial for differentiating hESCs into specific ectodermal tissues.”

After: “Especially during ectodermal differentiation, the presence of CPP-P8 downregulated the expression of genes closely associated with CNS development (*DCX*, *NEUROD1*, *NOS2*, and *PAX3*) and upregulated the essential transcription factors for retinal gliogenesis and NOTCH signalling (*LHX2*, *NOTCH1*, and *DLK1*) (Supplementary Fig. 7). Therefore, inhibitory peptides could be beneficial for differentiating hPSCs into specific ectodermal tissues related to the retinal gliogenesis differentiation.”

10. One of the siRNAs in Supplementary Table 5 contains Ts instead of Us. Is this a typo?

Response: The dTdT is correct. Overhang dT does not seem to have an effect on target mRNA recognition and degradation. Furthermore, dTdT could protect siRNA from the endogenous ribonuclease. We hope this adequately addresses your concerns.

Reviewer #2 (Remarks to the Author):

The authors of the manuscript entitled “Role of UPF1-Lin28A interaction during early differentiation of human embryonic stem cells” report an interaction between LIN28 and UPF1 and claim that this interaction regulates the efficiency of NMD. They derive a peptide inhibitor proposed to interfere with the LIN28-UPF1 interaction. They further claim that Lin28-UPF1 interacts to regulate the differentiation potential and pluripotency of hESCs.

The manuscript reports a potentially interesting finding, but quite bluntly put, neither experiments nor conclusions are up to the standard to be expected from a Nature Communications paper. Many experiments fail to exclude potential artefacts stemming from overexpression of Lin28 (or Upf1). The expression of RNAseq data is selective and incomplete and clear evidence for a functional role for the interaction of Lin28 and Upf1 is lacking. Crucial data are misinterpreted and there is no evidence for differential UPF1 activity by LIN28 regulation in hESCs. Further, analysis of hESC differentiation potential is crude and I am at a loss to see which conclusions can be drawn from the presented experiments.

Overall, the manuscript falls short on providing convincing evidence for a functionally important interaction between Lin28 and Upf1 involved in NMD and pluripotency. It would need considerable extra work and much more careful interpretation of results to perform the experiments needed to test the hypothesis underlying the claims made in this manuscript.

Detailed points:

1. The central claim that Lin28 overexpression reduces pUpf1 levels and thereby NMD activity is not supported by the data shown in Fig 3d. Reduced pUpf1 levels apparent on the top two panes follow the reduced actin levels in the bottom pane. This is a crucial experiment that does not support a central claim in the paper.

Response: We partially agree that the decreased p-UPF1 in Fig. 3d (in the original manuscript, now Supplementary Fig. 3a) might be due to less loading amount. However, we ascertain that the p-UPF1 was reduced due to overexpression of LIN28A, since the levels of UPF1 were comparable and the levels of β -actin were not entirely different.

To exclude the possibility that loading lower amounts resulted in a reduction of p-UPF1s, we tested the function of LIN28A on p-UPF1 by depleting the endogenous LIN28A in hPSCs and measured the levels of p-UPF1 using three biological replicates (Fig. 3c in the revised manuscript). In contrast to

the decreased p-UPF1 by overexpression of LIN28A in HeLa cells, depletion of LIN28A significantly increased the levels of p-UPF1 (pSer1096 and pThr28) in all three hPSCs types, suggesting that LIN28A regulates UPF1 phosphorylation. Moreover, we treated hPSCs with CPP-P8 (Supplementary Fig. 5b in the revised manuscript). Our findings reveal that the disruption of UPF1-LIN28A interaction induced by CPP-P8 for 3 days resulted in an upregulation of p-UPF1 in CHA15 and Pro2 cells, as illustrated in Supplementary Figure 5b. Interestingly, it is worth noting that CPP-P8 did not elicit a corresponding increase in p-UPF1 levels in H9 cells, even though the depletion of LIN28A was found to upregulate p-UPF1 in this context. This observed disparity might be attributed to a temporal delay in the manifestation of CPP-P8's effects. We hope these reproducible results in hPSCs are acceptable.

2. In general, it is hard to understand the author's choice of model cell lines. The abstract and title are about human embryonic stem cells, but only a few experiments have been performed in these cells. I have problems understanding why experiments were not performed in hESCs in the first place. hESCs are amenable for genetic modification and biochemistry. Hence, I really can't understand why not choose them for all experiments. As is, there is little evidence that an interaction between Lin28 and Upf1 exists in hES cells.

Response: The concerns raised in this regard are identical to those of other reviewers. The reasons we employed immortalized cells that did not constitutively express LIN28A for the mechanistic study instead of hPSCs were i) they are easy to handle for gene overexpression, and ii) the distinct effects of LIN28A function. Moreover, we determined the UPF1-LIN28A interaction using PLA assay in hPSCs and the LIN28A-regulatory effects on NMD in hPSCs (Fig. 1e and Fig. 6a in the original manuscript). However, we do agree that the mechanistic effects of LIN28A in hPSCs should be confirmed. To determine the UPF1 and LIN28A interaction in hPSCs, we reperformed IP experiments using anti-UPF1 antibody in hPSCs instead of IP using LIN28A-overexpressing in LIN28A-deficient cells, indicating that UPF1 and LIN28A form a complex in a partial RNA-independent manner (Supplementary Fig. 1c in the revised manuscript). Also, to confirm the effect of LIN28A on the UPF1-UPF2 interaction, we performed IP experiments in LIN28A knockdown hPSCs, demonstrating that UPF1 interacts with LIN28A and the depletion of LIN28A upregulated the UPF1-UPF2 interaction (Fig. 3b in the revised manuscript), which was consistent with observations using IP involving LIN28A overexpression. Furthermore, we performed a PLA assay using additional hPSCs, including H9 and Pro2, suggesting that UPF1 and LIN28A interact within the cells (Fig. 1e in the revised manuscript). Finally, the depletion of LIN28A in

hPSCs increased p-UPF1, suggesting that LIN28A also has a regulatory function on NMD (Fig. 3c in the revised manuscript). Thus, we claim that UPF1 interacts with LIN28A in hESCs, regulating NMD.

3. Levels of Lin28 overexpression are unclear in Fig2 and many others. It is crucial to test how levels compare with physiological levels? In general, the reliance on overexpression to show an interaction between Upf1 and Lin28a is prone to artefacts – this applies to multiple Figures in the manuscript. The authors made no effort to ensure expression at physiologically meaningful levels. Hence, it is possible that observed interactions by co-IP or PLA are artefacts of overexpression. To show an interaction in a physiologically meaningful context an interaction should be observed in a setting that relies on endogenous levels of protein, and ideally in the cell line (hESCs) that the authors want to study.

Response: Thank you for your valuable comments. We do agree with the reviewer's concern that the level of LIN28A expression was equivalent to the expression of LIN28A in hPSCs. Before submission, we had obtained the appropriate amount of DNA plasmid to express FLAG-LIN28A and GST-LIN28A. To compare the levels of overexpressed LIN28A in cell lines with the ones of LIN28A in hPSCs, we transfected FLAG-LIN28A in HeLa and 293T cells that were employed in this manuscript and performed WB (Supplementary Fig. 1b in the revised manuscript). The WB results demonstrated that the levels of exogenous FLAG-LIN28A in cell lines were comparable with the endogenous LIN28A in hPSCs, suggesting that overexpression of LIN28A in HeLa and 293T could reflect hPSCs. As we applied the PLA assay using PA-1 cells and hESCs in the original manuscript, we maintain that the results were not prone to artefacts. Furthermore, we performed the PLA assay using more hPSCs to detect an

endogenous UPF1-LIN28A interaction (Fig. 1e in the revised manuscript), indicating that the UPF1-LIN28A interaction was observed in all three hPSCs. We hope our explanation is acceptable.

4. Furthermore, the claim that the proposed Lin28-Upf1 interaction is independent of nucleic acids is not supported by the data. To conclude the Upf-Lin28a interactions are indeed independent of nucleic acids, the authors would need to perform co-IP in cells in their absence, e.g. by performing colIPs in the presence of benzonase or cynase.

Response: Thank you for the constructive suggestion. To determine whether UPF1 and LIN28A bind directly, we performed an IP experiment in the presence of nuclease (RNase, DNase, and MNase), *in vitro* binding, immunofluorescence, PLA, and deletional mutant assays by IP in the original manuscript (Fig. 1a, 1b, 1c, 1d, 1e, 4b, Supplemental Fig. 1a, 1b, 1c, 2b, and 2c in the original manuscript). However, we agree with these suggestions and reperformed the IP experiments in Fig. 1a, 1b (overexpressing LIN28A in 293T cells), and Supplementary Fig. 1c (hPSCs) in the revised manuscript in the presence of benzonase. First, we tested the degradation of nucleotide by benzonase (Supplementary Fig. 1a in the revised manuscript). Then, we analyzed the colPed FLAG-LIN28A with GST-MYC-UPF1 or colPed MYC-UPF1 with GST-LIN28A, respectively, using WB, which indicated that UPF1 and LIN28A partially interacted in an RNA-independent manner. These observations were confirmed when IP was performed in hPSCs. IP against UPF1-antibodies in the presence of benzonase indicated that the endogenous UPF1 and LIN28A interacted in an RNA-independent manner in hPSCs (Supplementary Fig. 1c in the revised manuscript). We hope our additional IP experiments using benzonase are acceptable.

Not available

5. There is no evidence for the statement “transcripts regulated by UPF1 and Lin28A were involved in the differentiation, development, and migration of stem cells.” I cannot see any specific regulation of stem cell associated genes in Fig 6b. Without any information on the significance at the individual gene level, this analysis remains meaningless.

Response: We agree with the reviewer’s point. In the original manuscript, we obtained the transcriptome analysis from a single experiment and RNA-seq. Thus, we repeated the experiment and RNA-seq using UPF1- or LIN28A-depleted hPSCs (Fig. 6a, 6b, and Supplementary Fig. 4b in the original manuscript) and the inhibitory peptide treated hPSCs (Fig. 6c, 6d, 6e, 7a, and Supplementary Fig. 4f in the original manuscript). We then reanalyzed the results using the transcripts that were significantly changed ($p < 0.05$). Consistent with Fig. 6a in the original manuscript, CDF graphs indicated that the depletion of UPF1 and LIN28A derepressed and repressed hPSCs NMD targets, respectively (Fig. 2e in the revised manuscript). However, as the reviewer noted, Fig. 6b (original manuscript) did not directly inform of the UPF1 effects on stem-cell associated transcripts. Therefore, we replaced the quadrant graph with a Venn diagram to show the transcripts commonly regulated by UPF1, LIN28A, and CPP-P8 using additional RNA-seqs (Fig. 6b in the revised manuscript). Consistent with Fig. 6c in the original manuscript, treatment of P8 significantly repressed hPSC NMD targets, suggesting that the disruption of UPF1-LIN28A interaction by CPP-P8 repressed hPSC NMD targets (Fig. 6a in the revised manuscript). Reanalysis using significantly changed transcripts demonstrated that 230 transcripts were commonly regulated over 1.2-fold, which were enriched in cellular development and differentiation (Fig. 6b and Supplementary Table 2, 3 in the revised manuscript). We replaced the revised RNA-seq analysis using statistically significant transcripts with the one in the original manuscript. We hope the reviewer accepts our explanation and results.

Original Manuscript

Revised Manuscript

Fig. 6a

Fig. 2e

Fig. 6b

Fig. 6b

Fig. 6c

Fig. 6a

6. The variance for most control measurements is not shown as their values were set to one. This gives the false impression of very robust levels in controls. This needs to be rectified and error bars should also be shown for the controls. The authors are also not clear about how statistical analysis was performed. Did they use the original measured values of the values normalized to one in comparison to controls, if the latter, the results are invalid.

Response: Thank you for your valuable suggestion. We set the normalized control value to 1.0, and then we measured the levels of the indicated transcripts in the original manuscript. That is why there is no variance in the normalized control values. As the reviewer suggested, instead of using the original measured values of the values normalized to one in comparison with controls, we obtained the normalized transcript values and then all the data values were normalized by the control average and the significance of the effects was retested. We replaced all our statistical analysis with the new test as described above (we displayed the one example as below). We hope the reviewer accepts our explanation and recalculation.

7. How is the claim in line 132 that Lin28 preferentially binds to unphosphorylated Upf1 supported by the data. Phosphorylated Upf1 has not been tested in this assay, so no comparison between Upf1 and pUpf1 can be drawn. Similarly, reduced colP of Upf1 with Lin28 after OA treatment is, at best, only

circumstantial evidence for a preferential binding of Lin28 to Upf1.

Response: Thank you for your valuable input. We discovered that LIN28A blocked UPF1-UPF2 binding (Fig. 3a in the original manuscript), suggesting that phosphorylated UPF1 decreased because UPF1 has been known to be phosphorylated after forming a UPF1-UPF2 complex.

After speculation of Fig. 3 results in the original manuscript, according to the reviewer's comment, we concluded that we could not give direct evidence that LIN28A prefers to bind to the unphosphorylated UPF1. Rather than preference binding of LIN28A to the unphosphorylated UPF1, LIN28A reduced the phosphorylated UPF1 by blocking the UPF1-UPF2 interaction; consequently, LIN28A interacted more with unphosphorylated UPF1 than phosphorylated UPF1. Similarly, the depletion of kinase (SMG1 or ATM1), which increased unphosphorylated UPF1, and OA treatment, which increased phosphorylated UPF1, may modulate the opportunity for unphosphorylated UPF1-LIN28A interactions (Fig. 3b and 3c in the original manuscript). Thus, we removed Fig. 3b and Fig. 3c in the original manuscript and performed additional experiments to determine the p-UPF1 by depletion of LIN28A or CPP-P8 treatment in hPSCs (Fig. 3b and Supplementary Fig. 5b in the revised manuscript, addressed in concern #1 raised by this reviewer). We apologize for this confusion and hope these modifications are acceptable.

8. The significance of RNAseq data remains unclear. The authors should provide clear data showing statistically significant changes in gene expression. How many individual NMD targets (or stem cell associated genes) show a significantly altered expression profile? How many other genes show a significant change. Is the enrichment of NMD targets significant? How exactly was ensured that the selection of random genes does not produce an artefact? Have multiple selection of 284 genes been performed in a bootstrapping approach to make sure that the choice of the author's 284 random genes was not a random 'lucky' pick? If so, how many permutations have been performed? Did the authors ensure that the random selected genes are expressed to a similar level as proposed NMD targets? The tables provided are not helpful and clearly below the expected reporting standard.

Response: Thank you for your constructive suggestion. We addressed the above concerns in #5; we obtained the statistically significant changed transcripts ($|F_c| > 1.2$, $p < 0.05$) via biological replicates. The 96, 60, and 51 NMD targets were significantly changed by UPF1-depletion, LIN28A-depletion, and P8-treatment in hESCs, respectively. Excluding NMD targets, the levels of 2321, 1991, and 2196 genes were significantly changed by UPF1-depletion, LIN28A-depletion, and P8-treatment in hPSCs, respectively. Regarding the enrichment of NMD targets, 4.1%, 2.0%, and 2.3% of significantly changed genes were NMD targets. Since UPF1 and LIN28A has multiple functions as posttranscriptional regulators, transcriptome changes could have resulted from indirect effects of posttranscriptional regulation, and thus a small percentage of NMD targets was observed.

We employed random genes in Fig. 2e and Fig. 6a (original manuscript). As this reviewer noted, random genes could be an artefact, we therefore replaced random genes with all significantly differential genes (SDGs) ($p < 0.05$). Regarding the table, we have added the p-values of all listed transcripts (Supplementary Table. 2 in the revised manuscript). We hope our explanation and additional analysis were acceptable.

9. How can the authors be sure that the effect of the Upf1 P8 peptide is specifically blocking the interaction with Lin28? Isn't it possible that the same domain serves as blocking site for other interactors as well?

Response: Thank you for noting your concerns. As the reviewer pointed out, CPP-P8 may cause off-target effects by interacting with other proteins. In the original manuscript, to exclude the possibility that peptides could raise the off-target effects, we performed the following. First, CPP-P8 itself did not have any effect on the NMD reporter levels and P8 did not block NMD in the absence of UPF1 or LIN28A, suggesting that CPP-P8 could specifically inhibit NMD in the presence of UPF1 and LIN28A (Supplementary Fig. 3b and 3c in the original manuscript). Second, the amino acid sequence of CPP-P8 from UPF1 were rarely matched with those from other proteins, suggesting that CPP-P8 may only interact with UPF1-binding partners (Supplementary Table. 2 in the original manuscript). Furthermore, in the revised manuscript, to strengthen the minimal off-target effects of CPP-P8, we performed RNA-seq using CPP-P1 or CPP-P8 treatment CHA15 cells in the presence or absence of LIN28A, and we then analyzed the differential expression between control siRNA and siLIN28A group (Supplementary Fig. 6d in the revised manuscript). RNA-seq results indicated that 42 and 53 transcripts were upregulated and downregulated over 2-fold by CPP-P8, respectively, which was almost reversed by the depletion of LIN28A. Likewise, CPP-P8 significantly repressed NMD targets in the presence of LIN28A, which was reversed by the depletion of LIN28A. All these results suggest that the peptide did not have any effect on posttranscriptional regulation without LIN28A or UPF1. Although we cannot completely exclude the possibility that CPP-P8 may interact with other proteins, we maintain that only CPP-P8 has these effects on NMD in the presence of UPF1 and LIN28A.

Supplementary Fig. 5c, 5d

Not available

Supplementary Fig. 6d

Not available

10. The effect of CPP8 on hESCs is not characterized well enough. Indeed, Nanog expression is lost upon treatment, but what is the fate of these cells? Will they die (as expected after inhibition of Upf1 function)? In line with this, what is the fate of hESCs upon deletion of Lin28a and Upf1? Are these cells still ESCs?

Response: We appreciate this comment. This concern is similar to that of Reviewer #3 (concern #4). To determine the fate of hPSCs through treatment with P8, we grew the P1, P2, and P8-treated hPSCs for a long period (20 days) with a split every 4–6-days. As indicated in Supplementary Fig. 6b in the revised manuscript, we observed that some hPSCs seemed to lose stem cell shape and displayed a differentiated cell shape, suggesting that P8-treatment induced spontaneous differentiation and the loss of stem cell makers, *POU5F1* (OCT4) and *NANOG*. Notably, treatment of CPP-P8 during proliferation upregulated the levels of differentiated markers. However, depletion of LIN28A or UPF1 in hPSCs for 2 days did not show any significant effects on the levels of *NANOG* and *POU5F1* (reviewer only version) or on the shape of hPSCs (data not shown). We might attempt to make stable UPF1 or LIN28A-knockdown hPSCs lines using lentiviral shRNA or the CRISPR/Cas9 system to observe the effects of UPF1- and LIN28A-depletion over a long period, as the siRNA effects did not last more than 5 days. However, since we have knowledge that the depletion of posttranscriptional regulators in selected stable cell lines by lentivirus or CRISPR/Cas9 may cause changes to the whole transcriptome during the selection process, we developed the peptide system to determine the specific effects of the UPF1-

LIN28A interaction. We were still unable to provide a complete explanation for the discrepancy in which peptides and the depletion of UPF1 or LIN28A show different effects on *NANOG*, *POU5F1*, and the spontaneous differentiation, but we would suggest that the effects from depletion of each gene may be different from the disruption of protein-protein interaction or may be due to the different treatment periods. Most of all, the depletion of proteins could not represent a disruption of protein-protein interaction. We hope our explanation and additional results (Supplementary Fig. 6b in the revised manuscript) are acceptable.

Original Manuscript

Revised Manuscript

Supplementary Fig. 6b

Not available

Reviewer only

Not available

11. Are any of the transcriptional changes in the heatmaps shown in Fig 8a statistically significant?

Response: Thank you for your question (Fig. 8a, which was mentioned should be Fig. 7a in the original manuscript as there was no Fig. 8). We reperformed RNA-seq using biological replicate samples; i.e. we treated hESCs with the indicated peptide and induced three germ-layer differentiation. We then analyzed the heat-map results using only significantly changed transcripts ($p < 0.05$) and the significantly changed levels of transcripts are marked (Fig. 7a in the revised manuscript). We hope our additional analysis and answers were acceptable.

12. In Fig2b increased Lin28 levels coincide with reduced MS2-Upf1 levels. Did the authors take into account that these reduced Upf levels could contribute to reduced NMD activity?

Response: Thank you for your viewpoint. The reviewer's concern is that the levels of MS2-HA-UPF1 in the fourth to sixth lanes looked less than those of the others, which might affect the increased levels of *G1* mRNA. Basically, an increase in MS2-HA-UPF1 enhanced the degradation of reporter *G1* mRNA. Thus, although MS2-HA-UPF1 in the fourth lane looked less than those in the sixth lane, the levels of *G1* mRNA in the sixth lane were significantly increased, which was due to the increased amount of FLAG-LIN28A. Consequently, the level of *G1* mRNA in the third lane was more upregulated than the one in the second lane, suggesting that LIN28A inhibits mRNA decay although the MS2-UPF1 levels in the second and third lane were comparable. Furthermore, we provided other biological replicates WB results where the levels of MS2-HA-UPF1 in gradual FLAG-LIN28A expression were comparable and *G1* mRNA were gradually increased. We added these results as a "Reviewer Version Only". Thus, we claimed that the increased *G1* mRNA resulted from the gradual expression of LIN28A and were not a result of the reduced MS2-HA-UPF1. We hope our explanation is acceptable.

Minor:

1. the authors state that all Lin28 OE experiments were performed in Lin28 KO cells. How do NMD levels in Lin28 KO cells compare to WT cells?

Response: Thank you for your concern. We did not completely understand the reviewer's point. Regarding the NMD targets (NMD levels), the NMD target in LIN28A KO cells (we believe LIN28A KO

cells represent the immortalized cell lines that do not constitutively express LIN28A) are completely different from the ones in LIN28A-expressing cells (i.e., hPSCs). The NMD targets in immortalized cell lines (HeLa and 293T) and the hPSCs that were employed in this study are obtained from previous reports^{1, 2}. If the reviewer is asking about the NMD efficiency (NMD levels), we could not adequately address this concern because the cellular transcript levels involved in NMD are not identical between cell lines and hPSCs. However, many studies have reported on the function of NMD in both cell lines and ESCs, and we believe that NMD happens in both cell lines. We hope our explanation addresses this concern.

2. How do transcript profiles of CPP-P8 treatment and UPF1 or LIN28 KD correlate?

Response: Thank you for your constructive comment. We analyzed the significantly changed NMD target levels using RNA-seqs from i) UPF1-depleted hPSCs, ii) LIN28A-depleted hPSCs, and iii) P8-treated hPSCs, and indicated the commonly regulated transcripts in Fig. 6b (revised manuscript). RNA-seq analysis using significantly changed transcripts demonstrated that 230 transcripts were commonly regulated by 1.2-fold, which were enriched in cellular development and differentiation (Fig. 6b in the revised manuscript). Moreover, 75 and 76 transcripts were upregulated and downregulated upon UPF1-depletion and CPP-P8 treatment, respectively, (below figure, reviewer only version) and there were 18 common transcripts, suggesting that approximately 24% transcripts overlapped.

Reviewer #3 (Remarks to the Author):

As an important RNA monitoring mechanism in eukaryotic cells, nonsense-mediated mRNA decay (NMD) identifies and degrades mRNAs containing premature termination codons (PTCs) in the open reading frame, which is important for maintaining orderly genes expression system. This research identifies the interaction of LIN28A and UPF1 (an important regulator of NMD) and the effect of this complex on NMD. The authors developed a cell-penetrating peptide (CPP) based on the LIN28A-UPF1 interaction structure to inhibit the interaction between LIN28A and UPF1. The addition of CPP-8 in hESC effectively inhibited the binding of LIN28A to UPF1, regulated the expression of ectodermal markers during differentiation, and disrupted hESC pluripotency.

Major Points:

1. In Figure 1, although different exogenous protein methods were used in different cells to examine the interactions between LIN28A and UPF1 in an RNA-independent manner, it is strongly recommended to validate LIN28A-UPF1 interactions in hESCs with immunoprecipitation of endogenous proteins, and comparing the effects before and after RNAase addition.

Response: Thank you for the suggestion. This concern is similar to those raised by other reviewers. Throughout the original manuscript, we performed most mechanistic studies using immortalized cells, which did not constitutively express LIN28A. To verify our mechanistic model in hESCs, we reperformed PLA assay, WB, and IP experiments in hPSCs (Fig. 1e and Supplementary Fig. 1c in the revised manuscript). As this reviewer's suggestion, we tested endogenous UPF1 and LIN28A interactions in the presence or absence of benzonase (more stringent nuclease than RNase mixture used in the original manuscript) via IP using anti-UPF1 antibody in hPSCs (Supplementary Fig. 1c in the revised manuscript). IP and WB results demonstrated that UPF1 partially interacts with LIN28A in an RNA-independent manner. We hope our additional results will be acceptable.

2. A variety of cell lines, including PA-1, HeLa, and 293T, were used in the study to demonstrate the regulatory effects of LIN28A and UPF1 on NMD. However, it is likely that hESC lines have different contexts and mechanisms. Many of the molecular experiments in this paper were not repeated in 3 different hESC cell lines (Figure 2-3). Another possibility to consider is iPSCs.

Response: Thank you for the constructive suggestion. The reviewer's suggestion that our mechanistic findings should be verified in hPSCs because most of the mechanistic studies have been done using LIN28A-deficient cell lines. To determine whether LIN28A has these effects on the NMD, we overexpressed GST-LIN28A in HeLa cells and performed RNA-seqs (Fig. 2e in the original manuscript).

CDF analysis confirmed LIN28A upregulated the expression of NMD targets. This was verified in hPSCs, where RNA-seq was performed using UPF1- or LIN28A-depleted CHA15 cells and RT-qPCR analysis using UPF1- or LIN28A-depleted H9 and Pro2 cells (Fig. 2e and Supplementary Fig. 2a in the revised manuscript).

To prove our mechanistic studies using immortalized cell lines were reproducible in hPSCs, we repeated Fig. 3a and Fig. 3d in the original manuscript, which indicated that LIN28A hindered UPF1-UPF2 interaction and LIN28A reduced UPF1 phosphorylation, respectively. First, we performed IP using anti-UPF1 antibodies in LIN28A-depleted hPSCs and we observed the level of co-IPed UPF2 (Fig. 3b in the revised manuscript), indicating that depletion of LIN28A enhanced the UPF1-UPF2 interaction in the presence of nuclease. Second, we measured the phosphorylated UPF1 in the absence of LIN28A in hPSCs (Fig. 3c in the revised manuscript). In contrast to the downregulation of phosphorylated UPF1 in LIN28A-overexpressing 293T cells (Supplementary Fig. 3a in revised manuscript), the depletion of LIN28A in hPSCs enhanced UPF1 phosphorylation. All our findings using immortalized cell lines in the original manuscript is reproducible in hPSCs. Although we did not perform all experiments using hPSCs in Fig. 2 (original manuscript) as the reviewer suggests, the key mechanistic experiments were performed and added to the revised manuscript. We hope the reviewer finds our suggestion of the mechanistic model of UPF1 and LIN28A in hPSCs is now acceptable.

3. Conventional immunofluorescence assay is affected by exposure time, focal length, etc., which makes it unsuitable for quantitative analysis. The authors should use an advanced instrument that can be used for quantitative analysis (Figure 6f, 7b).

Response: Thank you for the comment. In addition to the immunofluorescence assay, we employed flow cytometry analysis to quantify the indicated proteins in Fig. 6f and Fig. 7b in the original manuscript (Fig. 6c and 7b, Supplementary Fig. 6a and 7f in the revised manuscript). Flow cytometry analysis demonstrated that CPP-P8 treatment reduced OCT4- and NANOG-positive cells during proliferation. Furthermore, SOX1, LHX2, and DLK1-positive cells were decreased in CPP-P8 treated CHA15 cells during ectodermal differentiation. We hope our additional verification is acceptable.

Original Manuscript

Revised Manuscript

Fig. 6c and Supplementary Fig. 6a

Not available

Fig. 7b and Supplementary Fig. 7f

Not available

4. In addition to identifying the differentiation markers, what are the effects of LIN28A-UPF1 interactions on the functional differentiation phenotypes?

Response: Thank you for the constructive suggestion. We observed that the levels of stem cell markers, *POU5F1* (OCT4) and *NANOG*, were reduced when UPF1 and LIN28A interaction was disrupted by the inhibitory peptide CPP-P8 (Fig. 6f in the original manuscript). To determine the effects of CPP-P8 during proliferation, we grew hPSCs with CPP-P1, P2, or P8 for long periods (approximately 20-days) with a split every 4~6-days. After CPP-P8 treatment, the cell phenotypes treated with CPP-P8 looked more like the spontaneous differentiated cells than the ones treated with CPP-P1 and CPP-P2 (Fig. 6c and Supplementary Fig. 6b in the revised manuscript). Notably, treatment of CPP-P8 during proliferation upregulated the levels of differentiated markers.

The final experiment to determine the functional phenotypes used treated human fetal neural stem cells (ventral midbrain (VM)-NSCs or cortex (CTX)-NSCs), which was stressed by H₂O₂, with the peptides (CPP-P1 or P8), and differentiation to neurons was initiated. The immunofluorescence results indicated that P8 treatment increased cell survival (DAPI staining) and upregulated dopaminergic neuron markers, Tyrosine hydroxylase (TH) and TUJ1 compared with CPP-P1 treatment. We are preparing another manuscript to show the effects of CPP-P8 on neuronal differentiation, we would like to show these results in a “Reviewer Only Version”. We hope our additional experiments have addressed the reviewer’s concern.

Original Manuscript	Revised Manuscript
Fig. 1e	Supplementary Fig. 6b
Not available	

Reviewer only

Not available

Minor points:

5. Results for quantitative analysis by Western Blot are recommended to be labeled with grayscale intensity values above the protein bands (e.g. Figure 2)

Response: Thank you for your suggestion. This suggestion is similar to that raised by Reviewer1 (major point 3). We believe “e.g. Figure 2” must be “Figure 3” because the WB results in Figure 2 are more distinct than the ones in Figure 3. We quantified the WB bands in Fig. 3a, 3b, and Fig. 3c with the mean and standard deviations in the revised manuscript. Quantification indicated that depletion of LIN28A in hPSCs enhanced UPF1-UPF2 interaction and increased the phosphorylated UPF1. Regarding Fig. 2b, the similar concern was raised by Reviewer 2. We provided other biological replicates WB results where the levels of MS2-HA-UPF1 in gradual FLAG-LIN28A expression were comparable and *Gf* mRNA were gradually increased. We added these results as a “Reviewer Version Only”. We hope this quantification adequately addresses the reviewer’s concerns.

Original Manuscript

Revised Manuscript

Reviewer only

Not available

Fig. 3a

Fig. 3a

Not available

Fig. 3d

Fig. 3c

6. The RT-qPCR results suffer from a problem of reproducibility and variability (especially supplementary Figure 5).

Response: We agree with the reviewer's point. We observed that some RT-qPCR results showed large standard deviations, most of which resulted from the samples during three germ layer differentiation. We could not clearly identify a reason for this, but we speculate that strong derivation of differentiation may mask the effects of the inhibitory peptide, resulting in different values. However, to clearly address the efficiency of CPP-P8 on differentiation, we added three more biological replicates and retested significance, which reduced variability. We hope our explanation is acceptable.

7. Many of the gene names, mRNA names, and protein names in the Figures do not follow the standard conventions.

Response: Thank you for your comment. We have corrected our nomenclature and followed the

standard conventions: i.e. capital letter for human proteins and italic capital letter for human transcripts. Additionally, we replaced protein names with gene symbols.

Reference

1. Colombo, M., Karousis, E.D., Bourquin, J., Bruggmann, R. & Mühlemann, O. Transcriptome-wide identification of NMD-targeted human mRNAs reveals extensive redundancy between SMG6- and SMG7-mediated degradation pathways. *Rna* **23**, 189-201 (2017).
2. Lou, C.H. *et al.* Nonsense-Mediated RNA Decay Influences Human Embryonic Stem Cell Fate. *Stem Cell Reports* **6**, 844-857 (2016).
3. de Melo, J. *et al.* Lhx2 Is an Essential Factor for Retinal Gliogenesis and Notch Signaling. *J Neurosci* **36**, 2391-2405 (2016).
4. Diacou, R. *et al.* Cell fate decisions, transcription factors and signaling during early retinal development. *Prog Retin Eye Res* **91**, 101093 (2022).

REVIEWER COMMENTS

Reviewer #1 (Remarks to the Author):

The authors have successfully addressed my concerns and I support publication of the revised manuscript.

Reviewer #2 (Remarks to the Author):

In the revised version some substantial additional data has been generated. Some of my points have been clarified. However, the new data raises some further questions. Overall, I still find many of the claims premature and not well supported by data.

Points below refer to the original points raised:

Ad 1)

-The authors now provide evidence that Lin28 KD results in higher pUpf1 levels. This is supporting their conclusions. However, the converse should also be true. Lin28 OE should reduce pUpf1. This is a claim the authors still make in the revised manuscript, which is still not conclusively shown in the data they provide.

The authors argue that similar levels of Upf1 loading are sufficient control. I disagree. This is not how loading controls work. If there is indeed a similar band strength for Upf1 and weaker bands for Actin (as it appears in the plot shown), then the conclusion could be that there is simply more UPF1 in the presence of high levels of LIN28. This says nothing about the resulting total levels of pUPF1, which will be the ones that are of functional relevance. The Western is not of a high enough technical quality to draw clear conclusions. Ideally such important quantitative statements should also be supported by quantitative Western analysis on multiple replicates, as the authors have now done in some of their new data.

Ad4)

-The authors now conclude (see line 269) that UPF1 and LIN28 interact in an RNA independent manner. This is not entirely supported by the data. Actually, most of the interaction appears to be RNA dependent and the interaction is clearly weakened by Benzonase. I suggest a more careful interpretation of the data without overstatement.

Ad 5)

-The Venn diagram now shown is much clearer. However, for clarification, it should be distinguished between genes that are upregulated and genes that are downregulated, based on the fact that the different treatments are expected to have different outcomes because they impact NMD in different ways.

-The plots shown, e.g., Fig 6a are rather difficult to interpret. In Fig 6a, it appears that ~1/3 of NMD

targets show an increase in expression upon CPP-P8 treatment. What is the explanation or this? Further, in Fig 6a, it appears as if NMD targets behave no different from all DEGs upon CPP-P8 treatment. What is the explanation for this?

-Depletion of Upf1 is expected to decrease NMD efficiency, deletion of LIN28a or addition of CPP-P8, according to what I understand is the authors' model, should have the opposite effect. The CPP-P8 effect should be the subset of the Lin28 depletion effect that is dependent on the Upf1-Lin28a interaction. Is this the case? A quick inspection of the supplemental Table shows that the majority of transcripts have the same directionality (up/downregulated) in all three conditions. I find this surprising. Is this expected?

-Similar to the Venn diagram in 6b for all genes, the overlap of NMD targets should be shown.

-The authors observe that only a small fraction of deregulated genes in hESCs are NMD targets. The authors explain this by other functions of Upf1 and Lin28 and secondary and tertiary deregulation, this is conclusive and in line with previous papers. But it remains unclear what fraction of NMD targets are deregulated; here a large fraction of expressed NMD targets should show a clear deregulation, especially in the case of Upf1 depletion. Can the authors clarify this?

Ad 9)

-I am confused about the discrepancy in the number of deregulated genes after CPP-P8 treatment in Fig 6b (>2000) and Supp Fig 6d (<100). Shouldn't the numbers be similar, both experiments apply CPP-P8 to hESCs. What is the explanation for this?

-What is the authors explanation for why Lin28 depletion reduces the impact of CPP-P8. If CPP-P8 blocks Lin28 from accessing Upf1, then depletion of Lin28 should have no additional effect.

Additional points:

-I have problems understanding the last statement in the discussion: "Nevertheless, it is crucial to acknowledge that the effects of CPP-P8 on differentiation in H9 and Pro2 cells may yield distinct phenotypic outcomes." This is rather problematic. If the authors have good reason to believe that the phenotypic outcome of CPP8-P8 treatment is distinct between hESC lines, this needs a mechanistic explanation. It should not be too difficult for the authors to perform a differentiation experiment using also H9 and Pro2 hESCs or iPSCs to clarify this matter.

-Related to this point, the fact that in H9 cells the authors observe a mild reduction instead of an increase in pUpf1 signal upon CPP-P8 treatment is surprising and raises doubts about the reliability of CPP-P8. Either the Lin28a-Upf1 interaction functions differently in H9 cells, or the CPP-P8, for whatever reason, has a cell line specific effect. This needs to be clarified, because either of the two options is a serious limitation of the study.

-Importantly, it is not clear how much the Lin28-Upf1 interaction indeed "regulates" differentiation (e.g., l309). I think a more careful and more factual interpretation would be that the interaction is required for proper proliferation and differentiation. Whether it is a direct regulative event (no evidence), or simply a mechanism that maintains cellular homeostasis (maybe more likely) so that regulative events can unfold remains unclear.

-L243: "remained at 50% of 0-day up to day..." is unclear. Do the authors mean day 0 (zero)?

-L 290: "Upf1 is involved in maintaining mESCs, as downregulation of UPF1 maintained pluripotency". I cannot understand the meaning of this sentence.

Reviewer #3 (Remarks to the Author):

I appreciate the work the authors have put into revising the manuscript, and responding to our queries. However, some minor concerns remain:

1. The coordinates on the flow cytometry plots are too small to see clearly. And no statistical analyses were provided.
2. Regarding the fold changes in Supplementary Figure 2a, many fold changes were denoted as zero. Is there an error here?
3. According to the authors, the interaction between LIN28A and UPF1 decreases significantly after the addition of Benzonase nuclease. Their conclusion is LIN28A partially interacts with UPF1 in an RNA-independent manner. This conclusion is logically confusing. Either the interaction is RNA-dependent, or it is independent.

REVIEWER COMMENTS

Reviewer #1 (Remarks to the Author):

The authors have successfully addressed my concerns and I support publication of the revised manuscript. (Response) We are grateful for the reviewer's constructive feedback and recommendations to enhance our manuscript.

Reviewer #2 (Remarks to the Author):

In the revised version some substantial additional data has been generated. Some of my points have been clarified. However, the new data raises some further questions. Overall, I still find many of the claims premature and not well supported by data.

Points below refer to the original points raised:

Ad 1)

-The authors now provide evidence that Lin28 KD results in higher pUpf1 levels. This is supporting their conclusions. However, the converse should also be true. Lin28 OE should reduce pUpf1. This is a claim the authors still make in the revised manuscript, which is still not conclusively shown in the data they provide. The authors argue that similar levels of Upf1 loading are sufficient control. I disagree. This is not how loading controls work. If there is indeed a similar band strength for Upf1 and weaker bands for Actin (as it appears in the plot shown), then the conclusion could be that there is simply more UPF1 in the presence of high levels of LIN28. This says nothing about the resulting total levels of pUPF1, which will be the ones that are of functional relevance. The Western is not a of a high enough technical quality to draw clear conclusions. Ideally such important quantitative statements should also be supported by quantitative Western analysis on multiple replicates, as the authors have now done in some of their new data.

(Response) In our initial revision, we presented quantitative Western blot analyses of LIN28A-depleted hPSCs (CHA15, H9, and Pro2) depicted in Figure 3b. These analyses indicated that the ablation of LIN28A enhances the phosphorylation levels of UPF1. This observed increase in p-UPF1 levels upon LIN28A depletion is consistent with the notion that overexpression of LIN28A suppresses p-UPF1 levels. In response to the reviewer's request, we carried out further experiments to overexpress LIN28A and quantitatively assess the concomitant downregulation of p-UPF1 (Supplementary Figure 3a). Consistent with our expectations, LIN28A overexpression resulted in a reduction of p-UPF1 levels to 40% (at Ser1096) and 71% (at Thr28) relative to the control. We hope that the additional experiments and their respective quantifications are acceptable.

Ad4)

-The authors now conclude (see line 269) that UPF1 and LIN28 interact in an RNA independent manner. This is not entirely supported by the data. Actually, most of the interaction appears to be RNA dependent and the interaction is clearly weakened by Benzonase. I suggest a more careful interpretation of the data without overstatement.

(Response) We surmise that the reviewer was referencing line 259, which is the introductory sentence of the Discussion section in our previously revised manuscript, rather than line 269. We agree with the reviewer's suggestion and have subsequently moderated our assertions throughout the manuscript. The modifications are as follows:

Before (line 259):

"UPF1 and LIN28A directly interact in an RNA-independent manner."

After (now at line 261):

"UPF1 and LIN28A partially interacts in an RNA-independent manner."

Additionally,

Before

"These results indicate that UPF1 directly interacts with LIN28A."

After (line 100)

"These results indicate that UPF1 partially interacts with LIN28A in an RNA-independent manner."

We hope the modification addresses the reviewer's concerns.

Ad 5)

-The plots shown, e.g., Fig 6a are rather difficult to interpret. In Fig 6a, it appears that ~1/3 of NMD targets show an increase in expression upon CPP-P8 treatment. What is the explanation or this?

Further, in Fig 6a, it appears as if NMD targets behave no different from all DEGs upon CPP-P8 treatment. What is the explanation for this?

(Response) In Fig. 6a, our data illustrated that i) CPP-P2, as a negative control, cannot regulate NMD targets and the gene expression patterns are similar to those treated with CPP-P1 (as indicated by red line in Fig. 6a) and ii) CPP-P8 regulates NMD efficiency, thereby deregulating the expression levels of numerous genes. Furthermore, NMD targets exhibit a significant downregulation enrichment by CPP-P8 (CPP-P8/P1 - NMD targets vs CPP-P8/P1 - all significantly differentially expressed genes (SDGs)). Notably, in order to exclude the artifacts, we employed SDGs as this reviewer suggested in the initial revision, instead of random transcripts as in our original manuscript.

The reviewer's mention of an increase in about 1/3 of NMD targets likely refers to a comparison between CPP-P8/P1 - NMD targets and CPP-P2/P1 - NMD targets. Rather, we conducted a comparison between NMD targets and SDGs. The rationale is that 6756 genes are exhibiting significant changes ($P < 0.05$) in expression levels due to CPP-P8 treatment, ranging from small to substantial fold changes. Given the complexity of the cellular events, the levels of many NMD targets were deregulated by CPP-P8 as well as other factors that were resulted from SDGs. Therefore, our focus was on genes under comparable conditions: SDGs and NMD targets.

In this context, we observed that approximately 6% of NMD targets increased, while approximately 38% remained unchanged, and the remaining 56% exhibited a significant decrease in expression levels by CPP-P8 (see below figure). Moreover, 34 NMD targets exhibited significant changes in expression levels by CPP-P2, whereas the group treated with CPP-P8 resulted in 124 NMD targets exhibiting significant changes, representing a nearly four-fold difference in the size of the groups. So, it is challenging to assert that the NMD targets deregulated by CPP-P2 may not the same ones deregulated by CPP-P8. Hence, we interpreted that NMD targets significantly decrease in response to CPP-P8 treatment, and we concluded that there is a correlation between UPF1-LIN28A interaction and NMD efficiency.

The reason why it was not seemed to show the differences upon CPP-P8 is that we employed only significantly changed transcripts (SDGs) to analyze. For example, the downregulated NMD targets by CPP-P8 were more repressed than the downregulated SDGs (56%), although we employed the smaller size of transcripts (124 vs 6756). Conversely, merely 6% of NMD targets by CPP-P8 were upregulated. Thus, we claimed that CPP-P8 efficiently repressed NMD targets.

-Depletion of Upf1 is expected to decrease NMD efficiency, deletion of LIN28a or addition of CPP-P8, according to what I understand is the authors' model, should have the opposite effect. The CPP-P8 effect should be the subset of the Lin28 depletion effect that is dependent on the Upf1-LIN28a interaction. Is this the case? A quick inspection of the supplemental Table shows that the majority of transcripts have the same directionality (up/downregulated) in all three conditions. I find this surprising. Is this expected?

(Response) We appreciate your insightful suggestion. We expected the majority of transcripts could exhibit the same direction of expression. This anticipation stems from our consideration of the impact of UPF1-LIN28A interaction, which plays a role in regulating the overall efficiency level of the RNA surveillance system, NMD, thereby potentially affecting the entire transcriptome within the cell. Therefore, we have designed the experiments in Figure 6 to figure out whether the inhibition of the UPF1-LIN28A interaction regulates NMD pathway, and subsequently the overall transcriptome changes and their outcomes in hESCs.

Depletion of UPF1 or LIN28A and transcriptome analysis would inform us the transcriptome changes by both UPF1-LIN28A interaction-dependent and -independent regulations. In Fig. 6b, 852 (230+622) transcripts were directly or indirectly deregulated by depletion of UPF1 and LIN28A, suggesting that part of 852 transcripts could be deregulated by UPF1-LIN28A interaction. Then, we combined the significantly changed transcripts by CPP-P8 to define UPF-LIN28A interaction-mediated transcriptome. Indeed, 230 transcripts (approximately 27%) were commonly deregulated by UPF1, LIN28A, and CPP-P8, which were resulted from change of NMD efficiency and its subsequent consequences.

In the context of our proposed model, wherein the UPF1-LIN28A interaction influences NMD efficiency, leading to transcriptomic shifts, the up/down/down trend highlighted by the reviewer represents a transitional phase of this process, specifically denoting NMD efficiency (data for this is already presented in Fig. 2e and 6a). The 230 common regulated genes we observed can be considered as the outcome of the final stage, which is transcriptomic change. Thus, considering that the methods to increase/decrease of NMD efficiency may differ among the three groups but ultimately all involve inhibiting UPF1-LIN28A interaction, the fact that most regulated transcripts show a similar direction (up- or down-regulation) lends further credibility to our analytical approach and conclusions. We trust our elucidation addresses your concerns satisfactorily.

-Similar to the Venn diagram in 6b for all genes, the overlap of NMD targets should be shown.

(Response) Thank you for your suggestion. We provided the Venn diagram in Supplementary Fig. 6a. The reason the numbers of NMD targets were low is that we employed very stringently defined NMD targets (286 transcripts)¹ in hESCs. 96, 60, and 51 NMD targets were regulated by UPF1-depletion, LIN28A-depletion, and CPP-P8 in hESCs, respectively ($|F_c| > 1.2$, $p < 0.05$). And 10 NMD targets were commonly regulated.

Revised Manuscript

Supplementary Fig. 6a

-The Venn diagram now shown is much clearer. However, for clarification, it should be distinguished between genes that are upregulated and genes that are downregulated, based on the fact that the different treatments are expected to have different outcomes because they impact NMD in different ways.

-The authors observe that only a small fraction of deregulated genes in hESCs are NMD targets. The authors explain this by other functions of Upf1 and Lin28 and secondary and tertiary deregulation, this is conclusive and in line with previous papers. But it remains unclear what fraction of NMD targets are deregulated; here a large fraction of expressed NMD targets should show a clear deregulation, especially in the case of Upf1 depletion. Can the authors clarify this?

(Response) We appreciate your valuable comments. First, we would like to address the above concerns together as below. The modulation of the NMD pathway can ultimately induce significant changes in the stem cell transcriptome, which should be distinguished from mere regulation of NMD targets. Thus, we aimed to substantiate the physiological outcomes of UPF1-LIN28A interaction through NMD in cellular contexts, focusing on (i) the regulation of UPF1-LIN28A interaction, (ii) the regulation of NMD pathway efficiency, and (iii) the axis leading to transcriptome changes. NMD not only functions as an RNA surveillance mechanism but also intricately regulates gene expression by controlling various transcription factors and master regulators, thus influencing overall gene expression within cells. Hence, the majority of the genes in the Venn diagram in Fig. 6b are not directly impacted by NMD and thus the up-/down-pattern is not of primary importance. Furthermore, we employed a highly stringent criterion, utilizing a small set of NMD targets as mentioned the above concern. Thus, we do not think the addresses regarding the first concern will give much information. However, we provided the Venn diagram, that reviewer requested, as "reviewer version only".

The NMD targets we employed¹ were chosen based on a spectrum of parameters such as UPF1 knockdown-induced upregulated genes, transcript stability using transcription inhibitor, downstream exon-junction (dEJ), 3' untranslated region (3'UTR), and upstream open reading frame (uORF) in hESC. Despite this stringent criterion, we observed significant up-/down-regulation enrichment in the expression patterns of many NMD targets under UPF1 knockdown, LIN28A knockdown, and CPP-P8 treatment, providing high confidence in the relationship between UPF1-LIN28A interaction and NMD efficiency (Fig. 2e and Fig. 6a).

Throughout this analysis, we noticed that the large number of disrupter (siUPF1, siLIN28A, and CPP-P8)-specific genes that did not overlap with any other group suggested that each disrupter exerts various derivative regulations beyond the proposed axis. Moreover, given our use of a highly reliable but small-sized NMD target pool, we could anticipate a limitation where not all transcripts would be commonly regulated by the NMD pathway under all three conditions. Nevertheless, the impact of UPF1-LIN28A interaction on the NMD pathway is evident, and through gene expression patterns in stem cells involving UPF1-LIN28A interactions, we could define physiological outcomes.

To verify the effects of UPF1-depletion, LIN28A-depletion, and CPP-P8 treatment on NMD, we provided transcriptome analysis using significantly deregulated NMD targets using stringently defined NMD targets (286 transcripts) ($|F_c| > 1.2$, $P < 0.05$) as below. 71 transcripts were upregulated by UPF1-depletion, while 51 and 32 transcripts were downregulated by LIN28A-depletion and CPP-P8 treatment, suggesting that NMD targets were clearly deregulated (Supplementary Fig. 6a). We hope our description adequately addresses the reviewer's concerns.

Revised Manuscript

Supplementary Fig. 6a

reviewer version only

Ad 9)

-I am confused about the discrepancy in the number of deregulated genes after CPP-P8 treatment in Fig 6b (>2000) and Supp Fig 6d (<100). Shouldn't the numbers be similar, both experiments apply CPP-P8 to hESCs. What is the explanation for this?

(Response) We would like to extend our apologies for the inadvertent error related to the fold-change value presented in Supplementary Fig. 6d of our original manuscript. The previously stated value was 1.5-fold, which, upon re-evaluation, should be corrected to 2.0-fold. The discrepancy between the numbers presented in Fig. 6b and Supplementary Fig. 6d, as highlighted by the reviewer, stems from the different fold-change thresholds applied. As indicated in the figure legend, we employed significantly change transcripts over 1.2-fold and 2.0-fold in Fig. 6b and Supplementary Fig. 6d, respectively. Because over 2000 dots were visually too distracting, we represented over approximately 100 transcripts (over 2-fold) in Supplementary Fig. 6d. We trust that this clarification regarding the variation in transcript numbers influenced by CPP-P8 will address and alleviate the reviewer's concerns.

-What is the authors explanation for why Lin28 depletion reduces the impact of CPP-P8. If CPP-P8 blocks Lin28 from accessing Upf1, then depletion of Lin28 should have no additional effect.

(Response) Because we designed the inhibitory peptide (P8) supposed to bind to LIN28A, P8 reduced coimmunoprecipitated UPF1 with LIN28A, augmented NMD efficiency (Fig. 5c), decreased the UPF1-LIN28A interaction in cells (Fig. 5d), and increased p-UPF1 (Supplementary Fig. 5b). In order to observe these effects was dependent on LIN28A, we treated CHA15 cells with CPP-P8 in the presence or absence of LIN28A (Supplementary Fig. 6d). CPP-P8 significantly upregulated 42 transcripts and downregulated 53 transcripts over 2-fold in the presence of LIN28A. However, deregulated transcripts by CPP-P8 in the absence were dramatically decreased, suggesting that CPP-P8 worked to regulate transcripts in the presence of LIN28A. However, 14 transcripts (12 upregulated and 2 downregulated transcripts) were still deregulated by CPP-P8 in the absence of LIN28A, which may come from the residual LIN28A that could not be completely removed by siRNA. Approximately 40% of endogenous LIN28A was observed in RNA-seq analysis. Consequently, 6 out of 12 and 1 out of 2 transcripts deregulated by CPP-P8 in the absence of LIN28A were also observed in CPP-P8 treatment in the presence of LIN28A, suggesting that only 6 upregulated and 1 downregulated transcript over 2.0-fold were deregulated by CPP-P8 in the absence of LIN28A. The small fraction of deregulated transcripts, 7.4% (7 (6+1) out of 95 (42+53)), was observed in a LIN28A-independent manner. We hope we adequately addresses the reviewer's concerns.

Additional points:

-I have problems understanding the last statement in the discussion: "Nevertheless, it is crucial to

acknowledge that the effects of CPP-P8 on differentiation in H9 and Pro2 cells may yield distinct phenotypic outcomes.” This is rather problematic. If the authors have good reason to believe that the phenotypic outcome of CPP8-P8 treatment is distinct between hESC lines, this needs a mechanistic explanation. It should not be too difficult for the authors to perform a differentiation experiment using also H9 and Pro2 hESCs or iPSCs to clarify this matter.

(Response) We observed UPF1-LIN28A interaction and CPP-P8 effects in all hPSCs (CHA15, H9, and Pro2 cells) during proliferation and the results from three different cell lines were comparable through the manuscript. However, we cannot completely exclude the possibility H9 and Pro2 cells may exhibit different phenotypic outcomes. That is why we mentioned “Nevertheless, it is crucial to acknowledge that the effects of CPP-P8 on differentiation in H9 and Pro2 cells may yield distinct phenotypic outcomes.” in the Discussion section. However, we do agree with the reviewer’s concerns and the sentence would not be beneficial to the manuscript. Thus, we removed the misleading sentence. We hope our explanation is acceptable.

-Related to this point, the fact that in H9 cells the authors observe a mild reduction instead of an increase in pUpf1 signal upon CPP-P8 treatment is surprising and raises doubts about the reliability of CPP-P8. Either the Lin28a-Upf1 interaction functions differently in H9 cells, or the CPP-P8, for whatever reason, has a cell line specific effect. This needs to be clarified, because either of the two options is a serious limitation of the study.

(Response) We examined the effects of CPP-P8 on UPF1 phosphorylation (p-UPF1) (Supplementary Fig. 5b). In this experiment, we treated each hPSC with CPP-P8 for two days and then, WB was performed to evaluate the levels of p-UPF1. As the reviewer pointed, CPP-P8 efficiently upregulated p-UPF1 levels in CHA15 and Pro2 cells, but not in H9. We described this discrepancy as followings in the first-round revision; Interestingly, it is worth noting that CPP-P8 did not elicit a corresponding increase in p-UPF1 levels in H9 cells, even though the depletion of LIN28A was found to upregulate p-UPF1 in this context. This observed disparity might be attributed to a temporal delay in the manifestation of CPP-P8’s effects. So, we employed the longer treatment of CPP-P8 in H9. After 4-day treatment of CPP-P8 in H9 cells and p-UPF1 levels were evaluated by WB, suggesting that longer treatment upregulates the levels of p-UPF1 as those in CHA15 and Pro2 cells. We exchanged the figures and described the incubation period in figure legend.

Since the first report of human embryonic stem cells (hESCs) in 1998, many kinds of ESC lines have been established, and scientists have conducted comparative analyses of their properties. Consequently, it was discovered that the characteristics of proliferation and differentiation varied among the cell lines^{2, 3}. Extensive analysis has been undertaken since the establishment of hiPSCs to determine whether or not the characteristics of hESCs and hiPSCs are comparable⁴⁻¹⁰. Some studies on differentiation experiments have shown that the differentiation potential and efficiency of iPSC lines and ESC lines can be different even under the same differentiation conditions¹¹⁻¹³.

It is widely acknowledged that while hESCs and hiPSCs share similar characteristics as hPSCs, there are numerous distinctions among individual cell lines. Even within sublines derived from a single cell line, differentiation characteristics can vary, according to a recent study published in *Cell Stem Cell*⁴. The variations in properties exhibited by these cell lines not only represent a significant obstacle to stem cell research but also complicate the interpretation of the data. In this study, we acknowledge completely that hESCs and hiPSCs differ in a number of distinguishing characteristics. Nevertheless, these variations among cell lines are prevalent in the field of stem cell research, and we believe it is appropriate to analyze the overall trend when interpreting the findings. In the future, we believe that a greater number of cell lines should be utilized to conduct comprehensive analyses. We hope the additional experiment and explanation will now satisfy the reviewer’s concerns.

Original Manuscript	Revised Manuscript
Supplementary Fig. 5b	Supplementary Fig. 5b

-Importantly, it is not clear how much the Lin28-Upf1 interaction indeed “regulates” differentiation (e.g., l309). I think a more careful and more factual interpretation would be that the interaction is required for proper proliferation and differentiation. Whether it is a direct regulative event (no evidence), or simply a mechanism that maintains cellular homeostasis (maybe more likely) so that regulative events can unfold remains unclear. (Response) Thank you for your valuable comments. In Discussion section, we broadly described what we observed. However, our description was ambiguous. We agree with the reviewer’s points and changed the sentence as below. We hope our description is now acceptable.

Before

“Exogenous LIN28A expression in differentiated cells directly interacts with UPF1, inhibiting the formation of UPF1-UPF2 complex and NMD. In hPSCs, the endogenous LIN28A-UPF1 complex reduces NMD efficiency and regulates cell differentiation and proliferation.”

After

“Exogenous LIN28A expression in differentiated cells directly interacts with UPF1, inhibiting the formation of UPF1-UPF2 complex and NMD. The endogenous LIN28A-UPF1 complex reduces NMD efficiency in hPSCs, thereby maintaining the stemness of proliferating cells.”

-L243: “remained at 50% of 0-day up to day...” is unclear. Do the authors mean day 0 (zero)?

(Response) We apologize the inaccurate description. What we intended to describe “0-day” is the day we initiated differentiation. We changed the sentence as below.

Before

“UPF1 and LIN28A levels remained at 50% of 0-day up to day 5, indicating that UPF1-LIN28A interaction exists in the early differentiation period.”

After

“UPF1 and LIN28A levels remained at 50% of their pre-differentiation baseline over five days post-differentiation, indicating that UPF1-LIN28A interaction exists in the early differentiation period.”

-L 290: “Upf1 is involved in maintaining mESCs, as downregulation of UPF1 maintained pluripotency.”. I cannot understand the meaning of this sentence.

(Response) We apologize the unclear description. We changed the sentence as below. We hope our modification is now clear.

Before

“Upf1 is involved in maintaining mESCs, as downregulation of UPF1 maintained pluripotency”

After

“Depletion of UPF1 maintained pluripotency, indicating the involvement of UPF1 in the regulation of proliferation and differentiation within mESCs”

Reviewer #3 (Remarks to the Author):

I appreciate the work the authors have put into revising the manuscript, and responding to our queries. However, some minor concerns remain:

1. The coordinates on the flow cytometry plots are too small to see clearly. And no statistical analyses were provided.

(Response) We appreciate the feedback provided. In response to the reviewer's suggestions, we have enhanced the clarity of the raw flow cytometry data presented in Supplementary Figures 6a and 7f by magnifying the indicated images. The flow cytometry plots in Supplementary Figures 6a and 7f are representative, and the quantification of relative positive cells is depicted in Figures 6c and 7d. Statistical analyses pertaining to these figures can be found in the revised manuscript. We trust that these modifications sufficiently address the reviewer's concerns.

2. Regarding the fold changes in Supplementary Figure 2a, many fold changes were denoted as zero. Is there an error here?

(Response) The values denoted in black within Supplementary Figure 2a approached zero. For enhanced clarity, we transitioned from a linear scale to a logarithmic scale as indicated. We trust that this adjustment satisfactorily addresses the reviewer's concerns.

3. According to the authors, the interaction between LIN28A and UPF1 decreases significantly after the addition of Benzonase nuclease. Their conclusion is LIN28A partially interacts with UPF1 in an RNA-independent manner. This conclusion is logically confusing. Either the interaction is RNA-dependent, or it is independent.

(Response) independent manner. This conclusion is logically confusing. Either the interaction is RNA-dependent, or it is independent.

(Response) The well-known RNA-binding proteins, UPF1 and LIN28A, engage in direct interactions with RNA, modulating RNA abundance accordingly. Given this characteristic, UPF1 and LIN28A may either coexist mediated by RNA or directly interact in the absence of RNA. This dual interaction potential is encapsulated in our description as "a partial RNA-dependent manner". Notably, immunoprecipitation (IP) assays presented in Figures 1a, 1b, and Supplementary Figure 1c demonstrate a reduction in the UPF1-LIN28A interaction following nuclease treatment. In the context of this research, we devised a CPP-peptide to selectively disrupt the direct UPF1-LIN28A interaction, omitting the RNA-mediated complex. Our findings underscore the

indispensability of this direct interaction in maintaining stem cell properties. Additionally, during the first-round revision, Reviewer 1 commented, "A stronger interaction in the absence of nucleases wouldn't necessarily negate the study's significance." We trust that this comprehensive explanation sufficiently addresses the reviewer's remarks.

References

1. Lou, C.H. *et al.* Nonsense-Mediated RNA Decay Influences Human Embryonic Stem Cell Fate. *Stem Cell Reports* **6**, 844-857 (2016).
2. Osafune, K. *et al.* Marked differences in differentiation propensity among human embryonic stem cell lines. *Nat Biotechnol* **26**, 313-315 (2008).
3. Chen, A.E. *et al.* Optimal timing of inner cell mass isolation increases the efficiency of human embryonic stem cell derivation and allows generation of sibling cell lines. *Cell Stem Cell* **4**, 103-106 (2009).
4. Parrotta, E. *et al.* Two sides of the same coin? Unraveling subtle differences between human embryonic and induced pluripotent stem cells by Raman spectroscopy. *Stem Cell Res Ther* **8**, 271 (2017).
5. Chin, M.H., Pellegrini, M., Plath, K. & Lowry, W.E. Molecular analyses of human induced pluripotent stem cells and embryonic stem cells. *Cell Stem Cell* **7**, 263-269 (2010).
6. Kim, K. *et al.* Epigenetic memory in induced pluripotent stem cells. *Nature* **467**, 285-290 (2010).
7. Bar-Nur, O., Russ, H.A., Efrat, S. & Benvenisty, N. Epigenetic memory and preferential lineage-specific differentiation in induced pluripotent stem cells derived from human pancreatic islet beta cells. *Cell Stem Cell* **9**, 17-23 (2011).
8. Bock, C. *et al.* Reference Maps of human ES and iPS cell variation enable high-throughput characterization of pluripotent cell lines. *Cell* **144**, 439-452 (2011).
9. Hussein, S.M. *et al.* Copy number variation and selection during reprogramming to pluripotency. *Nature* **471**, 58-62 (2011).
10. Shao, K. *et al.* Induced pluripotent mesenchymal stromal cell clones retain donor-derived differences in DNA methylation profiles. *Mol Ther* **21**, 240-250 (2013).
11. Mills, J.A. *et al.* Clonal genetic and hematopoietic heterogeneity among human-induced pluripotent stem cell lines. *Blood* **122**, 2047-2051 (2013).
12. Hu, B.Y. *et al.* Neural differentiation of human induced pluripotent stem cells follows developmental principles but with variable potency. *Proc Natl Acad Sci U S A* **107**, 4335-4340 (2010).
13. Kajiwara, M. *et al.* Donor-dependent variations in hepatic differentiation from human-induced pluripotent stem cells. *Proc Natl Acad Sci U S A* **109**, 12538-12543 (2012).
14. Pantazis, C.B. *et al.* A reference human induced pluripotent stem cell line for large-scale collaborative studies. *Cell Stem Cell* **29**, 1685-1702.e1622 (2022).

REVIEWERS' COMMENTS

Reviewer #2 (Remarks to the Author):

The authors have addressed my concerns. I recommend to be very clear in the final version of this manuscript that that the depletion and inhibitor treatments on the known set of NMD targets have the expected effect in terms of directionality of gene expression. This has only been properly shown in the second round of revision and earlier clarification would have alleviated many concerns in the first place.

Reviewer #3 (Remarks to the Author):

No further comments, I am satisfied with the authors' responses.

REVIEWER COMMENTS

Reviewer #2 (Remarks to the Author):

The authors have addressed my concerns. I recommend to be very clear in the final version of this manuscript that the depletion and inhibitor treatments on the known set of NMD targets have the expected effect in terms of directionality of gene expression. This has only been properly shown in the second round of revision and earlier clarification would have alleviated many concerns in the first place. (Response) We appreciate your comments. In response to your comments, we have provided a more detailed description of the directionality of gene expression in NMD targets upon UPF1 depletion, LIN28A depletion, or CPP-P8 treatment in the Results and the Discussion sections. We hope our description adequately addresses the reviewer's concerns.

Results section (Line 225):

"RNA-seqs using UPF1-depleted, LIN28A-depleted, or CPP-P8 treated hPSCs indicated that 230 transcripts including 10 NMD targets were commonly regulated, which were enriched in cellular development and differentiation, suggesting that regulation of NMD targets potentially affects the entire transcriptome."

Discussion section (Line 317):

"In the context of our proposed model, wherein the UPF1-LIN28A interaction influences NMD efficiency, leading to transcriptomic alterations, particularly evident in NMD efficiency (Fig. 2e and Fig. 6a). The 230 transcripts commonly regulated can be considered as the outcome of the final stage, which is transcriptomic change (Fig. 6b). Thus, considering that the methods to increase/decrease of NMD efficiency by UPF1-depletion, LIN28A-depletion, and CPP-P8 treatment may differ among the three groups but ultimately all involve inhibiting UPF1-LIN28A interaction, the fact that most regulated transcripts show a similar direction (Supplementary table 2)"

Reviewer #3 (Remarks to the Author):

No further comments, I am satisfied with the authors' responses.

(Response) We express gratitude for the constructive feedback and valuable recommendations provided by the reviewer to improve our manuscript.